# Priming of dendritic cells by DNA-containing extracellular vesicles from activated T cells through antigen-driven contacts

Daniel Torralba [1,2], Francesc Baixauli [1,3], Carolina Villarroya-Beltri[1,2], Irene Fernández-Delgado[1,2], Ana Latorre-Pellicer[4], Rebeca Acín-Pérez[5], Noa B Martín-Cófreces [1,2,6], Ángel Luis Jaso-Tamame[7], Salvador Iborra [5], Inmaculada Jorge[6,8], Gloria González-Aseguinolaza[9], Johan Garaude [5], Miguel Vicente-Manzanares [10], José Antonio Enríquez [5,11], María Mittelbrunn[12,13] & Francisco Sánchez-Madrid[1,2,6]

Interaction of T cell with antigen-bearing dendritic cells (DC) results in T cell activation, but whether this interaction has physiological consequences on DC function is largely unexplored. Here we show that when antigen-bearing DCs contact T cells, DCs initiate antipathogenic programs. Signals of this interaction are transmitted from the T cell to the DC, through extracellular vesicles (EV) that contain genomic and mitochondrial DNA, to induce antiviral responses via the cGAS/STING cytosolic DNA-sensing pathway and expression of IRF3-dependent interferon regulated genes. Moreover, EV-treated DCs are more resistant to subsequent viral infections. In summary, our results show that T cells prime DCs through the transfer of exosomal DNA, supporting a specific role for antigen-dependent contacts in conferring protection to DCs against pathogen infection. The reciprocal communication between innate and adaptive immune cells thus allow efficacious responses to unknown threats.

[1] Vascular Pathophysiology Research Area, Centro Nacional Investigaciones Cardiovasculares (CNIC), 28029 Madrid, Spain. [2] Servicio de Inmunología, Instituto Investigación Sanitaria Princesa, Universidad Autónoma de Madrid, Diego de León 62, 28006 Madrid, Spain. [3] Immunometabolism Department, Max Planck Institute for Immunobiology and Epigenetics, 79108 Freiburg im Breisgau, Germany. [4] Grupo de Medicina Xenómica, CIBERER, Universidad de Santiago de Compostela, 15782 Santiago de Compostela, Spain. [5] Myocardial Pathophysiology Research Area, Centro Nacional Investigaciones Cardiovasculares (CNIC), 28029 Madrid, Spain. [6] Centro de Investigación Biomédica en Red Enfermedades Cardiovasculares (CIBERCV), Melchor Fernández Almagro 3, 28029 Madrid, Spain. [7] MRC Clinical Sciences Centre, Imperial College Faculty of Medicine, London SW7 2AZ, UK. [8] Laboratory of Cardiovascular Proteomics, Centro Nacional Investigaciones Cardiovasculares (CNIC), 28029 Madrid, Spain. [9] Centro de Investigación Médica Aplicada (CIMA), Universidad de Navarra, 31008 Pamplona, Spain. [10] Instituto de Biología Molecular y Celular del Cáncer USAL-CSIC, 37007 Salamanca, Spain. [11] Centro de Investigaciones en RED (CIBERFES), Melchor Fernández Almagro 9, 28029 Madrid, Spain. [12] Instituto de Investigación Sanitaria, Hospital 12 de Octubre (i+12), 28041 Madrid, Spain. [13] Centro de Biología Molecular, UAM-CSIC, Departamento de Biología Celular e Inflamación, 28049 Madrid, Spain. These authors contributed equally: Daniel Torralba, Francesc Baixauli. Correspondence and requests for materials should be addressed to F.S-M. (email: fsmadrid@salud.madrid.org)

The generation of a specific immune response against a pathogen requires the initial interaction of an antigen-specific T cell with antigen-presenting cells (APCs), specifically dendritic cells (DCs). DCs express antigenic peptides associated to the major histocompatibility complex (MHC) class II, and T cell recognition of these complexes leads to the formation of a stable intercellular junction between the cells, the immune synapse (IS)[1]. The IS comprises a highly organized dynamic supramolecular structure that orchestrates the early events of T cell activation, including the spatiotemporal organization of the TCR signaling and its costimulatory molecules. The IS is a cell polarization event involving the redistribution of membrane-associated receptors and the cytoskeleton, and the polarization of intracellular trafficking and secretory organelles[2]. IS formation generates diverse regulatory checkpoints for the control of antigen-specific T cell response by controlling the spatial and temporal rearrangements of the different T cell receptors and organelles.

Besides its prominent role in instructing T cell activation, the IS transmits intracellular signals that direct DC function[3]. On the DC side, IS formation rapidly increases the concentration of MHC class II molecules in the synaptic contact to strengthen antigen presentation to cognate T cells[4]. The actin and microtubule cytoskeletons of DCs undergo substantial rearrangements during IS formation, allowing the polarization of different compartments. The relocation of endosomal compartments in DCs mediates the polarized secretion of cytokines into the synaptic region[5,6], and the trafficking of the major histocompatibility complexes[4]. IS formation increases DCs survival by inhibition of apoptotic signaling, enhancing DC antigen presentation, and T cell clonal expansion[7]. The IS integrates the signaling provided by the APC and the T cell to modulate the function of both cells and ensure T cell priming, activation, and efficient T cell responses against cognate antigens. The underlying mechanisms and functions of the IS in T cells are widely known although the physiological significance of the IS for DCs is still largely unexplored.

The transfer of bioactive molecules from the T cell to the DC through the IS constitutes a main vehicle of intercellular communication. T cells and DCs exchange numerous molecules, including cytokines, membrane receptors, membrane patches, signaling molecules, or genetic material (mainly functional microRNAs) during IS formation[8]. Immune cells readily transfer membrane fragments to other cells, membrane-associated receptors, and co-receptors. Several cellular mechanisms mediate information transfer between the T cell and the DCs, including transendocytosis, trogocytosis, formation of tunneling nanotubes, and polarized secretion of extracellular vesicles (EVs)[8]. These exchanges fine-tune the activation of the T cell, e.g., by cell-extrinsic depletion of costimulatory proteins from APCs; capture of MHC class I and II from target cells; downregulation of TCR peptide-MHC II complexes from APCs; regulation of DCs gene expression, and transcellular signaling through the transfer of microRNA-loaded exosomes and TCR-enriched EVs[9–12]. Pathogens such as bacteria or virus subvert the architecture of the IS to enhance dissemination between immune cells[13–15]. The IS may act as a portal that supports the transfer of an array of different molecules or even entire pathogens between T cells and DCs.

Immune cells secrete a variety of EVs to the extracellular milieu that can exert diverse immune functions, including antigen presentation, immune activation, induction of tolerance, or suppression of immune responses[16]. EVs comprise apoptotic bodies, ectosomes or microvesicles, and exosomes[17,18]. Exosomes are distinguished by their unique endocytic origin. Exosomes form by invagination of the multivesicular body (MVB) membrane and are released to the extracellular medium upon the fusion of the MVB with the plasma membrane. Protein and lipid sorting and packaging into exosomes occur in a regulated manner involving mono-ubiquitination and the endosomal sorting complexes required for transport (ESCRT) machinery, association with lipid rafts or the tetraspanin network[19]. Exosomes contain proteins involved in their biogenesis, vesicle trafficking, lipid membrane organization, as well as proteins and adhesion receptors specific of the producing cell. Exosomes are particularly enriched in genetic material, mostly RNA species such as small RNAs and long non-coding RNAs, as well as DNA[19]. This makes them attractive candidates to mediate cell-to-cell communication. However, the ability of the genetic material contained within exosomes to evoke immune signaling responses in recipient cells is largely unexplored.

In the present manuscript this question is investigated in the context of the immune cognate interactions. Data show that T cell EV contain genomic and mitochondrial DNA (mtDNA) endowed with the ability to trigger immune signaling in recipient cells. By using different proteomic, genomic, and cell biology approaches, it is demonstrated that the unidirectional transfer of DNA within EVs from T cells to DCs triggers antiviral responses in the DC. Such DNA-based priming confers DCs resistance to subsequent viral infections. In conclusion, resulting data illustrate a way by which T cells promote an alert state in DCs that protects them against posterior infection driven by the vesicular transfer of DNA.

## Results

**T cells shuttle genomic and mitochondrial DNA in exosomes.** EVs transferred from T cell to APC contain many different types of biologically active molecules, including proteins and genetic material[9,10]. To investigate the possible specific function of the different components of the biologic material transferred from the T cell to the DC, the protein and genetic content of EVs isolated from the culture supernatant of primary T lymphoblasts were characterized by differential ultracentrifugation[20] (Fig. 1a). Diameters of EVs obtained in the $100,000 \times g$ fraction ranged from 50 to 200 nm (Fig. 1b), in agreement with the reported size of exosomes[17]. Proteomic characterization of T cell EVs identified abundant peptides derived from canonical exosome components, including tetraspanins, endosomal proteins, cytoskeletal proteins, Rab-related trafficking proteins, heat-shock proteins, and nucleotide-binding proteins (Supplementary Data 1). Gene Ontology (GO) term enrichment analysis revealed that a major fraction of the total number of identified peptides (>3000 peptides in four biological replicates) were significantly associated with the nuclear and mitochondrial fractions (Fig. 1c and Supplementary Data 2). Mitochondrial peptides constituted $7.5 \pm 0.3\%$ of the total exosome proteome, consistent with proteomic studies on exosomes isolated from other cell types, including immune cells[21,22]. EVs isolated from resting T lymphoblasts and cells activated with a polyclonal stimulus barely contained proteins of the mitochondrial matrix and the inner or outer mitochondrial membranes (Fig. 1d). EVs from activated T lymphoblasts contained significant amounts of mitochondrial transcription factor A (TFAM), a highly abundant mtDNA-binding protein, and the mitochondrial nucleoid-forming protein ATPase family AAA-Domain containing protein 3 (ATAD3)[23] (Fig. 1d). The presence of the canonical exosomal proteins CD81 and TSG101 in the purified $100,000 \times g$ fraction suggested an endosomal origin (Fig. 1d). TFAM and ATAD3 co-fractionated with the exosomal proteins CD81 and TSG101 in the $100,000 \times g$ EV fraction from sucrose-gradient isolation, (Fig. 1e), localizing mtDNA-binding proteins to exosomes.

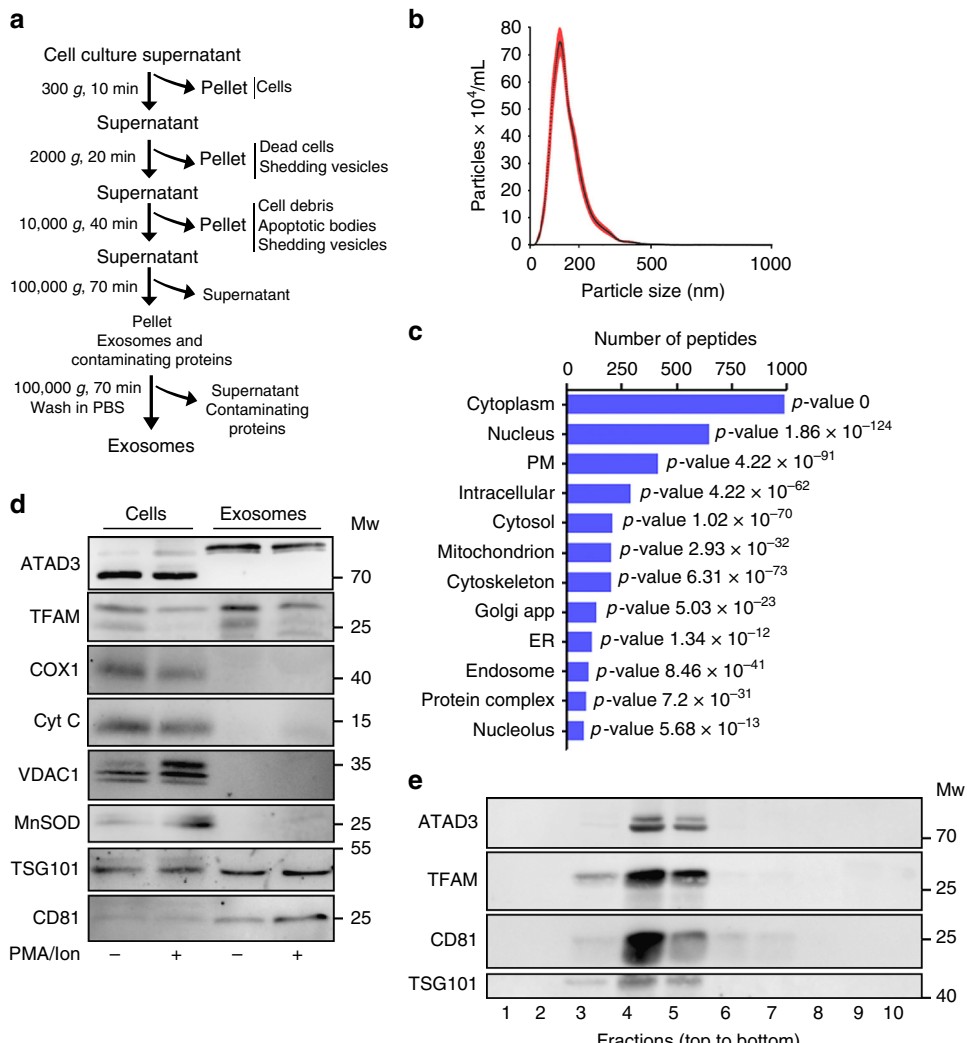

**Fig. 1** Exosomes shuttle mitochondrial proteins. **a** Differential ultracentrifugation protocol for isolating and purifying exosomes from cell culture supernatants. **b** Size distribution analysis by Nanoparticle Tracking Analysis (NTA) of purified EVs from primary human T lymphoblasts. **c** Gene ontology (GO) cellular component analysis of peptides identified in T cell EVs. **d** Western blot analysis of mitochondrial proteins in EVs obtained from the culture supernatant of primary human T lymphoblasts with or without treatment with phorbol myristate acetate (PMA) plus ionomycin. Cells and EVs were blotted for proteins associated with mtDNA (TFAM and ATAD3), for proteins located in the inner mitochondrial membrane (COX1 and Cytochrome C), the outer mitochondrial membrane (VDAC1), the mitochondrial matrix (mitochondrial manganese superoxide dismutase, MnSOD), and for the exosome markers TSG101 and CD81. **e** Western blot analysis of ATAD3, TFAM, and the exosome markers CD81 and TSG101 in sucrose fractions. EVs obtained from the culture supernatant of human T lymphoblasts were laid on a discontinuous sucrose gradient and floated by overnight centrifugation. Gradient fractions were collected and analyzed by immunoblot to reveal the distribution of mtDNA-binding proteins and exosomal proteins in the sucrose fractions from lower to higher sucrose density (left to right). Gels shown are representative out of three independent experiments

Among EVs, exosomes are enriched in genetic material, mostly non-coding RNAs[24,25]. Exosomes from cancer cell lines and patients contain DNA that reflects the mutational status of the parental tumor cells[26,27]. Deep-sequencing analysis of DNA in the EVs secreted by primary T lymphoblasts identified several reads that covered nuclear sequences and the entire mitochondrial genome (Fig. 2a). These results were confirmed by polymerase chain reaction (PCR) amplification of indicated regions of the mitochondrial genome and the *B2M* nuclear gene from the DNA obtained from the $100,000 \times g$ fraction (Fig. 2b). The DNA content of the isolated EVs was reduced by DNase treatment whereas a small fraction of the DNA remained protected unless vesicles were treated with DNase and a lipid destabilizing agent (Fig. 2c). Two pools of DNA were present, one accessible (likely on the surface of the vesicle) and one inaccessible (likely in the lumen of the vesicle). The presence of

mtDNA in the $100,000 \times g$ EV fraction was confirmed by the partial co-fractionation on sucrose gradients with the exosomal proteins CD63 and TSG101 (Fig. 2d). Flow cytometry analysis of EVs coupled to aldehyde sulfate beads reveals that the majority of beads were positive for both CD81 and DNA (Fig. 2e, and Supplementary Fig. 1a). Total internal reflection fluorescence (TIRF) microscopy-based analysis of the co-localization of exosomal marker CD81 with TSG101 or with DNA in isolated EVs showed that CD81 fully co-localized with TSG101, but only partly with DNA (Supplementary Fig. 1b), suggesting the presence of DNA in a subset of CD81$^+$ exosomes in addition to other different populations of EVs.

DNA staining in non-permeabilized EVs increased upon permeabilization (Fig. 2e), confirming that DNA is present both outside and inside EVs. To examine the oxidative status of EV DNA, the 8-hydroxydeoxyguanosine (8-OHdG) DNA

modification was evaluated as a hallmark of DNA oxidation. Staining with anti-oxidized DNA antibody (Ab) is also detected in non-permeabilized EVs (Fig. 2f). In agreement with these results, chromatin immunoprecipitation of EV lysates with 8-OHdG Ab followed by DNA isolation and PCR analysis revealed the presence of several oxidized mtDNA and genomic DNA regions in EVs when compared with the control isotype Ab (Supplementary Fig. 1c). To address the preferential localization of oxidized mtDNA, EVs and mitochondrial fractions of the corresponding cells were isolated and chromatin immunoprecipitation (ChIP) was performed with the 8-OHdG and total DNA

Abs. The enrichment in oxidized mtDNA at the EV fraction was demonstrated by qPCR analysis of mitochondrial genes in ChIP and quantification of the ratio between oxidized DNA vs total DNA (Fig. 2f). These results indicate that T cells release EVs carrying genomic DNA and mtDNA and associated mtDNA-binding proteins, and that the released DNA is partially oxidized.

**Segregation of mtDNA and related proteins into MVB pathway.** The presence of mitochondrial components in EVs prompted us to investigate the molecular mechanisms controlling their extracellular release. We first co-transfected HEK293 cells

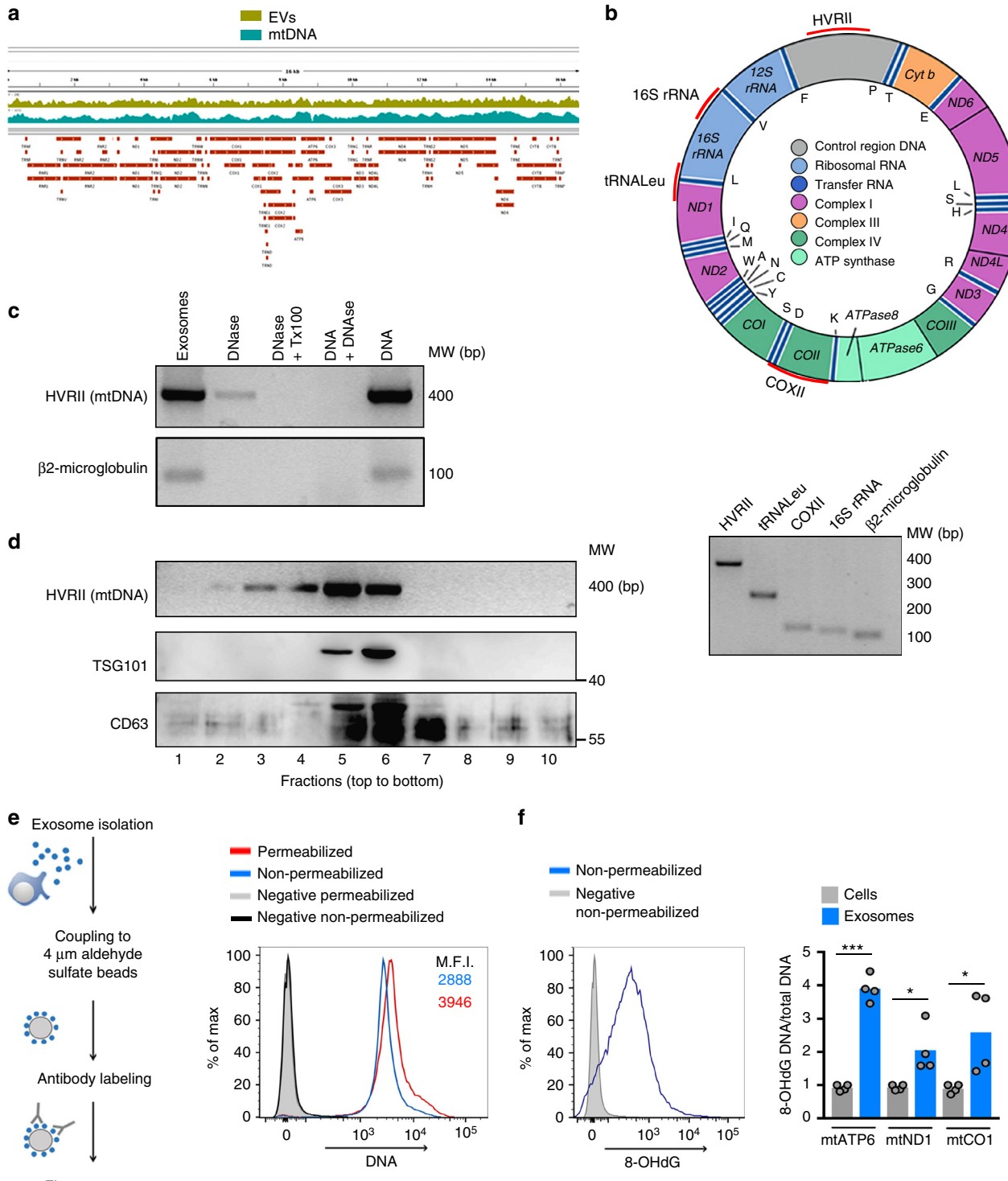

with TFAM-GFP and a mitochondria-targeted red fluorescent protein (mitoDsRed), then stained for endosomal markers. Confocal immunofluorescence analysis excluded co-localization of mitochondria and TFAM with the early endosome antigen 1 (EEA1), an early endosomal marker. Mitochondria and TFAM partially co-localized with components of maturing endosomal structures: HRS (hepatocyte growth factor-regulated tyrosine kinase substrate), a component of the ESCRT machinery; lyso-bisphosphatidic acid (LBPA) and ceramide-positive structures, which are markers of late endosomes and MVBs; and CD63, a highly abundant tetraspanin found in MVBs and exosomes (Supplementary Fig. 2a). To rule out potential analysis artifacts produced by the exogenous overexpression of TFAM-GFP, we measured the co-localization of endogenous mtDNA-binding protein single stranded DNA-binding protein 1 (SSBP1) with EEA1 and CD63 in HEK293 cells treated with low doses of FCCP, that promotes these events[28]. Increased co-localization of SSBP1 with CD63 indicates preferential localization in late endosomal structures (Fig. 3a). The segregation of mitochondrial components into maturing endosomal structures was confirmed in HEK293 cells transfected with constitutively active Rab7 (Q67L), which enhances endosomal trafficking, resulting in the formation of enlarged late endolysosomal structures[29]. Rab7 (Q67L) triggered the accumulation of endogenous TFAM within Rab7-positive endosomal structures, co-localizing with mitoDsRed (Fig. 3b). Also, immunogold labeling of TFAM and DNA revealed their appearance in canonical multivesicular body structures in electron microscopy studies (Supplementary Fig. 2b), overall supporting the involvement of the endosomal pathway in the secretion of mitochondrial components in EVs.

To further elucidate the nature of the vesicles containing mitochondrial components, we inhibited exosome biogenesis in T cells by targeting the neutral sphingomyelinase 2 (nSMase2, SMPD3) (Supplementary Fig. 3a). This enzyme participates in exosome biogenesis by triggering the budding of intraluminal vesicles into MVBs[30]. These conditions reduced the recovery of exosomal proteins as well as mtDNA-binding proteins and mtDNA in the purified $100,000 \times g$ fraction (Fig. 3c), showing that mtDNA and mtDNA-binding proteins are released through canonical exosomal pathway and not in other EVs such as apoptotic bodies. The blocking of exosome biogenesis by silencing nSMase2 or MVB fusion with the plasma membrane by targeting Rab27a with specific shRNAs (Supplementary Fig. 3a) increased mitochondrial mass, intracellular TFAM levels, intracellular ROS, and DNA oxidative damage (Fig. 3d). The suppression of nSMase2 or Rab27a altered mitochondrial morphology and

cristae organization, with increased mitochondrial cristae width (Fig. 3e). Changes in mitochondrial cristae organization relate to changes in mitochondrial electron transport chain complex organization that in turn may result in functional changes in the respiration ability of cells. Specifically, tight cristae provide more efficient mitochondrial OXPHOS[31], whereas loose cristae appear in highly glycolytic cells[32]. Consistent with a loose organization of mitochondrial cristae, Rab27a and nSMase2 silencing significantly decreased mitochondrial respiration (Fig. 3f). These findings indicate that mitochondrial function is altered in the absence of proper exosome biogenesis and secretion and suggest that the release of mitochondrial content in exosomes contributes to mitochondrial homeostasis and mtDNA metabolism.

We next investigated the contribution of mitochondrial turnover and degradation mechanisms to the secretion of mitochondrial components in EVs. Macroautophagy, from here on referred to as autophagy, is a self-degradation process that recycles cellular constituents, including misfolded or aggregated proteins and damaged organelles[33]. Confocal microscopy analysis did not reveal co-localization of autophagy proteins ATG5 and LC3 with MVBs or mitochondria (Supplementary Fig. 3b). Serum deprivation, a reported autophagy stimulus, decreased exosome secretion concomitantly with reduced extracellular release of mitochondrial components (Fig. 3g), consistent with the autophagy-promoted specific degradation of MVBs[34]. Moreover, inhibition of autophagosome cargo degradation by bafilomycin-A1-mediated blockade of autophagosome–lysosome fusion did neither alter the detection of TFAM in the exosomal fraction nor the concentration and size of the purified EVs (Fig. 3g and Supplementary Fig. 3c). This finding indicated that autophagosome secretion or content release upon bafilomycin-A1 treatment does not account for the presence of mitochondrial components in EVs. Additionally, specific siRNA silencing of the late autophagy mediator LC3 in J77 T cells did not alter exosome concentration and mitochondrial content (Fig. 3h and Supplementary Fig. 3d). Together, these findings indicate that the loading and secretion of mitochondrial material in EVs occurs independently of general macroautophagy.

**DNA and mitochondrial components transfer at immune contacts.** The presence of mitochondrial constituents in exosomes led us to hypothesize that mitochondrial material could be transferred from the T cell to the APC during the formation of antigen-dependent contacts. To assess this, we first evaluated whether isolated exosomes shuttle mitochondrial components

**Fig. 2** T cell exosomes shuttle mtDNA and genomic DNA. **a** Representation of the number of sequence reads aligned with the mitochondrial genome in EVs (EVs) and mitochondrial extracts (mtDNA) isolated from human primary T cells. **b** Upper panel: Diagram of the mitochondrial genome, indicating the mtDNA regions analyzed by PCR (marked in red: HVRII, tRNALeu, COXII, and 16 S rRNA). Lower panel: Agarose gel showing the amplification products of the different mtDNA regions and the genomic DNA (*β2-MICROGLOBULIN*) analyzed in EVs purified from primary human T cells. **c** Agarose gel electrophoresis showing the detection of DNA in purified EVs. EVs obtained from the culture supernatant of J77 T were treated with DNase or DNase and Tx-100. Levels of mtDNA and genomic DNA were assessed by PCR amplification (*HVRII* and *β2-MICROGLOBULIN*, respectively). Isolated DNA from J77 T cells treated or not with DNase (DNA + DNase and DNA, respectively) is shown as control for DNase activity. **d** Distribution of mtDNA and TSG101 and CD63 in sucrose fractions. EVs from human Jurkat T cells were laid on a discontinuous sucrose gradient and floated by overnight centrifugation. DNA from gradient fractions was PCR-amplified for the *HVRII* mtDNA region. Proteins from gradient fractions were analyzed by immunoblot for TSG101 and CD63. A representative gel is shown ($n = 3$). **e** Exosomes from primary mouse T lymphoblasts were purified, coupled to aldehyde sulfate beads, and analyzed by flow cytometry by staining with the anti-DNA Ab under permeabilizing and non-permeabilizing conditions. Histograms show DNA staining in exosomes. Numbers are mean fluorescence intensities of the positive population from a representative experiment ($n = 3$). **f** Left: Histogram shows oxidized DNA staining in exosomes coupled to aldehyde sulfate beads with anti-8-hydroxydeoxyguanosine (8-OHdG)Ab under non-permeabilizing conditions. Right: qPCR analysis showing the enrichment of the indicated mitochondrial genes in the DNA obtained from exosomes and mitochondrial fractions from the same exosome-producing cells. DNA from both fractions was immunoprecipitated with 8-OHdG and total DNA Abs. Graph shows the ratio between oxidized DNA and total DNA in exosomes relative to their content in the mitochondrial fractions from the producing cells in duplicates from two independent experiments. *t*-test: *$P$-value < 0.05, ***$P$-value < 0.0001

between cells. Purified exosomes from non-transfected J77 T cells or cells stably transfected with a mitochondria-targeted fluorescent protein, mitoDsRed, were added to Raji B lymphoblastoid cells, used as APCs. After incubation with exosomes from J77mitoDsRed cells, recipient cells exhibited mitoDsRed fluorescence (Fig. 4a). We next assessed whether exosomes can mediate intercellular mtDNA transfer. Donor and acceptor cell mtDNAs were distinguished by the use of conplastic mice with the same nuclear genome (C57BL/6) but different mtDNA haplotypes, from the C57BL/6 mice strain (C57[C57]) or from the NZB strain (C57[NZB])[35]. These two mtDNA haplotypes differ in 12 missense polymorphisms identifiable by restriction length fragment polymorphism (RFLP) analysis (Fig. 4b). Incubation of T cell-derived

exosomes from C57[C57] mice with DCs differentiated from the C57[NZB] mice resulted in uptake of C57[C57] mtDNA by recipient C57[NZB] DCs, as shown by the acquisition of the C57[C57] haplotype and the presence of C57[C57] RFLP fragments in exosome-treated DCs (Fig. 4b), supporting the transfer of mitochondrial components and mtDNA between cells through exosomes.

The mostly unidirectional nature of exosome transfer from T cells to DCs[9,10] suggested that the exchange of mitochondrial information between immune cells occurs upon antigen-dependent contacts. TCR Vβ8[+] J77 T cells expressing a mitochondria-targeted fluorescent protein (mitoYFP) or a fluorescent version of TFAM (TFAM-DsRed) were co-cultured with Raji B cells preloaded with *Staphylococcus enterotoxin*

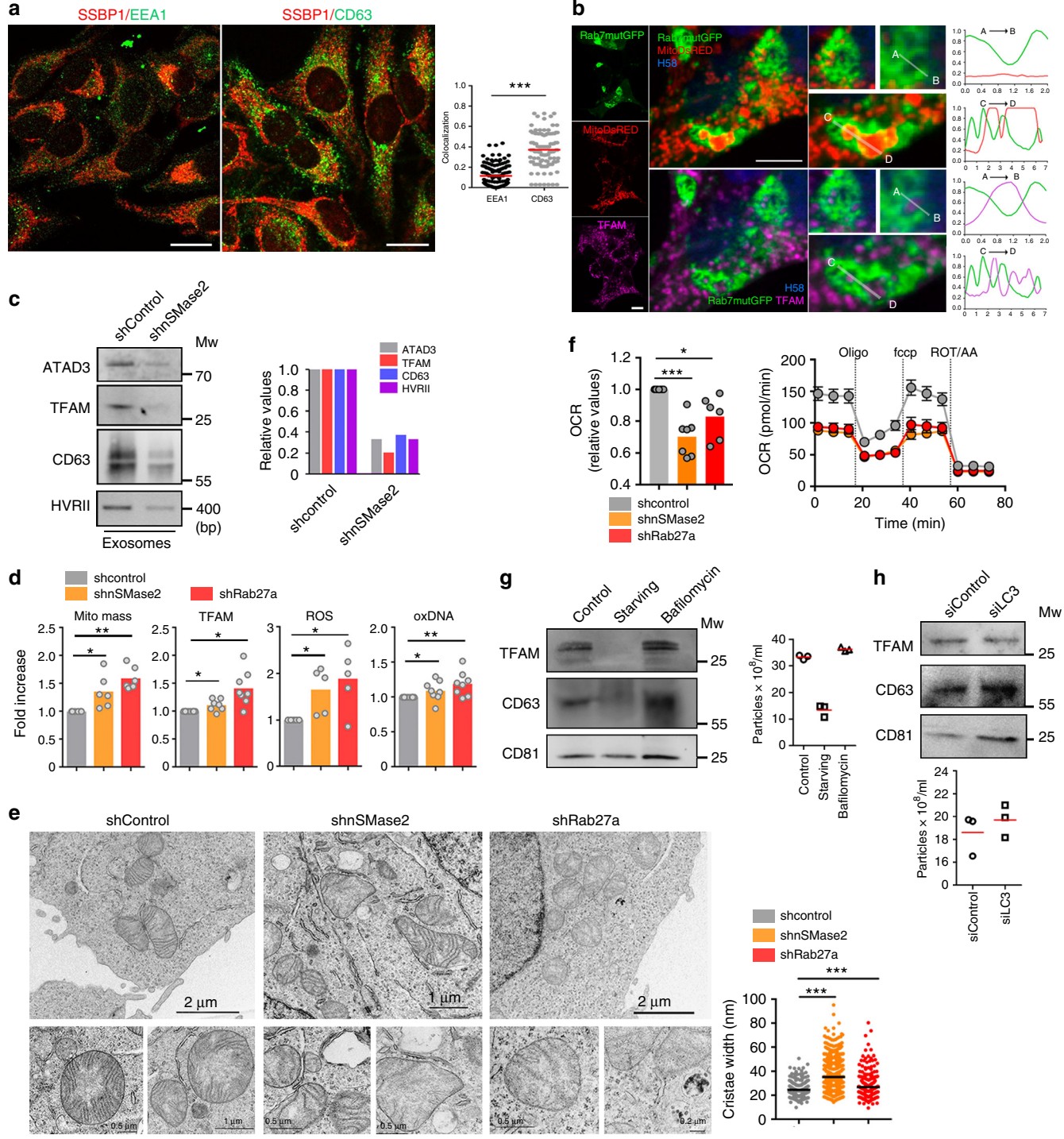

*superantigen-E* (SEE), which mimics antigen recognition when confronted with TCR Vβ8+ T cells. Flow cytometry analysis revealed transfer of mitochondria (mitoYFP) and TFAM (TFAM-DsRED) from T cells to APCs during antigen-dependent interactions (Fig. 4c). To address this in primary cells, we set up a model of antigenic presentation in which bone marrow-derived DC (BMDC) from C57NZB mice presented OVA peptide to CD4+ T cells from transgenic OT-II mice (C57C57), which are specific for that antigen (Fig. 4d)[35]. After 16 h co-culture in the presence or absence of OVA peptide, we detected CD4+ T cell mtDNA only in recipient APCs from co-cultures including OVA peptide antigen (Fig. 4e), indicating that intercellular mtDNA transfer requires antigenic triggering. Using CD4+ T cells from OT-II C57NZB mice and BMDCs from the C57C57 mice, we confirmed that mtDNA transfer occurred unidirectionally from T cells to BMDCs (Fig. 4f).

**cGAS/STING-IRF3 pathway senses DNA transfer by exosomes.** Mitochondrial components are a major source of danger-associated molecular patterns (DAMPs). They can trigger innate immune responses, likely due to their similarities to bacterial ancestors[36]. We hypothesized that the transfer of mitochondrial content and mtDNA during T cell-APC antigen-dependent contacts may represent a mode of cell-to-cell transmission of danger-associated messages. To examine this possibility, we interrogated which signaling pathways were controlled by T cell exosomes in recipient DCs. This was carried out by characterizing the changes to the gene profile of DCs upon incubation with T cell exosomes by RNA deep-sequencing analysis (Fig. 5a). Gene expression profiling identified more than 1600 significantly altered genes in recipient DCs exposed to T cell exosomes (Supplementary Fig. 4a, b). Exosomes regulated the expression of a major set of genes involved in interferon (IFN) type I responses and antiviral activity, including IFN-stimulated genes (ISGs) *Ifit1, Ifit2, Ifit3, Isg15, Usp18*, and *Cxcl10*, and the antiviral signaling factors *Gbp5* and *Gbp6* (Fig. 5a). Analysis of the genes that changed their expression profile in response to T cell exosomes by Gene Ontology annotation indicated they were mainly involved in IFN signaling, exogenous DNA and RNA sensing, bacteria and viruses recognition, and crosstalk between innate and adaptive immune cells (Fig. 5b). These findings suggested that T cell exosomes prime the antiviral innate immune response in DCs, creating a gene signature that upregulates antiviral and antimicrobial responses to control pathogen infection and replication (Fig. 5c). We validated the RNA sequencing results by PCR for the genes *Cxcl10, Isg15, Ifit1, Ifit3, Usp18*, and *Stat1* (Fig. 5d). One of the most relevant pathways involved in ISG expression depends on cGAS/STING interaction upon cytoplasmic detection of endogenous or exogenous DNA[37]. cGAS functions as a main sensor of viral and bacterial DNA in the cytoplasm of infected cells. Upon its activation by DNA detection, it generates the second messenger cyclic dinucleotide cGAMP, which binds to and activates STING. STING in turn activates TANK-binding Kinase 1 (TBK1), which phosphorylates interferon regulatory factor 3 (IRF3) to promote its translocation to the nucleus, where it induces the expression of IFNβ and ISGs. Triggering of cytoplasmic DNA sensors is associated with intracellular clustering of STING[38]. Consistent with activation of the cGAS/STING DNA-sensing pathway, exosomes induced a significant aggregation of STING in DCs (Fig. 5e). The expression of ISGs in response to exosomes was reduced in STING-deficient ($Sting^{Gt/Gt}$) DCs (Fig. 5d), and completely abrogated in $Irf3^{-/-}$ DCs (Fig. 5f). The role of exosomal DNA in triggering these responses was evaluated by treating exosomes with DNase. DNase treatment significantly reduced the ability of exosomes to trigger the activation of interferon-related genes in recipient WT DCs (Fig. 5g), suggesting that the DNA located in the outer surface of exosomes is important for full induction of antiviral responses in recipient cells. To ascertain whether DNA located into the lumen of exosomes may also contribute to antiviral responses in recipient DCs, we genetically engineered a plasmid expressing DNase II fused to the intraluminal domain of the exosome-enriched protein CD63. qPCR analysis of isolated exosomes showed that exosomes from cells stably expressing CD63-DNase II contain less mitochondrial DNA than exosomes from control cells stably expressing wild-type CD63 (Supplementary Fig. 4c). Exosomes from cells expressing CD63-DNase II triggered a lower antiviral response in recipient cells compared with exosomes obtained from CD63-expressing control cells (Supplementary Fig. 4d). Overall, these results support that DNA on the surface and the inside of T cell exosomes can act transcellularly, i.e., in recipient DCs, to prime antiviral responses through activation of the STING-IRF3 DNA-sensing pathway.

**Primed antiviral responses in DCs by synaptic T-exosomes.** We next assessed whether antigen-driven T-APC contacts trigger antiviral responses in recipient cells through the transfer of exosomal components. Gene expression analysis revealed that

**Fig. 3** Mitochondrial components segregate into MVBs and are secreted in exosomes. **a** Confocal co-localization analysis in HEK293 cells treated with FCCP (4 h) and stained with SSBP1 (red), EEA1 and CD63 (green). Bar, 10 μm. Graph shows means for Mander's co-localization coefficient for SSBP1 and EEA1 or CD63 ($n = 3$). t-test: ***P-value < 0.0001. **b** Confocal co-localization analysis in HEK293 cells co-transfected with Rab7-Q67L-GFP (Rab7mutGFP, green) and mitoDsRed (red), and immunostained for TFAM (purple) and nuclei (HOECHST 58, blue). Center and right images show high magnification of left images (Rab7-Q67L-GFP and mitoDsRed in upper panels; Rab7-Q67L-GFP and TFAM in lower panels). Charts: Fluorescence profiles along the corresponding white lines. Bar, 10 μm. **c** Mitochondrial components in the exosome fraction obtained from equal numbers of shControl and shnSMase2 J77 T cells. ATAD3, TFAM, and CD63 were detected by immunoblot; mtDNA was detected by PCR for *HVRII*. Graph: Quantification of exosomal proteins and mtDNA in a representative experiment ($n = 3$). **d** Flow cytometry analysis of mitochondrial mass (Mitotracker green), intracellular ROS levels (DCFDA staining), oxidized DNA (8-OHdG Ab staining), and endogenous TFAM in HEK293 cells knocked down for nSMase2 and Rab27a. Graphs: Quantification from 5–8 independent experiments (Mean). t-test *P-value < 0.05; **P-value < 0.001. **e** Electron microscopy images show defects in mitochondrial ultrastructure and cristae organization in shnSMase2 and shRab27a HEK293 cells. Graph: Quantification of mitochondrial cristae width (Mean). t-test: ***P-value < 0.0001. **f** Graph: Basal oxygen consumption rate (OCR) of control, shnSMase2 and shRab27a HEK293 cells. Dots represent mean from three independent experiments run in duplicate or triplicate. t-test, *P-value < 0.05, ***P-value < 0.0001. Chart: OCR from shControl, shnSMase2 and shRab27a HEK293 cells in response to oligomycin (Oligo), fccp, and rotenone plus antimycin A (Rot/AA). ($n = 2$; mean ± S.E.M.). **g** Western blot analysis of exosomes from Jurkat T cells left untreated, serum-starved overnight or treated with bafilomycin A. Membranes were blotted for TFAM, CD81, and CD63. Graph: Nanoparticle concentration in the exosomal fractions (mean, two independent experiments). **h** Western blot analysis of exosomes obtained from Jurkat T cells transfected with control or LC3 siRNA. Graph: Nanoparticle concentrations in the exosomal fractions (mean, $n = 2$). Western blots are representative out of three independent experiments

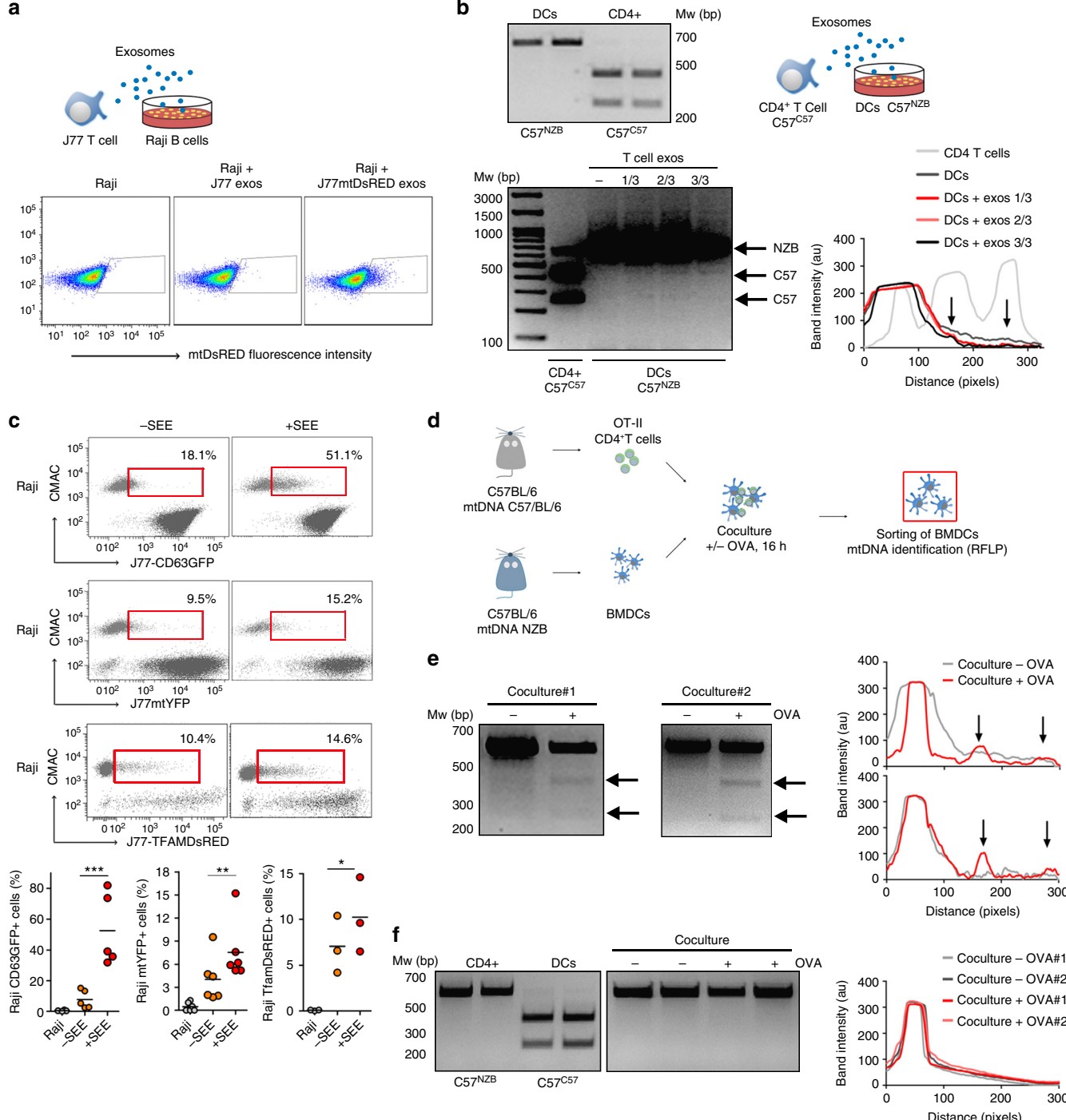

**Fig. 4** Transfer of mitochondrial components during immune cognate interactions. **a** Flow cytometry analysis of Raji B cells incubated overnight with exosomes from J77 T cells control or stably expressing mitoDsRed (Representative experiment, $n = 3$). **b** Up: RFLP analysis of the mitochondrial genomes of C57[C57] and C57[NZB] mice. Total DNA is from C57[C57] T lymphocytes and C57[NZB] BMDCs. C57[C57] haplotype includes a BamHI restriction site; enzyme digestion results in two smaller fragments of 414 pb and 250 pb. Digestion of PCR products from C57 [NZB] haplotype results in a single band of 664 pb (full length). Down, RFLP detection of exogenous mtDNA in C57[NZB] DCs incubated overnight with increasing amounts of exosomes from C57[C57] CD4+ T lymphoblasts. 414 pb and 250 pb fragments appear in C57[NZB] DCs upon addition of exosomes (lanes 1/3, 2/3, and 3/3). Chart: Intensity profile of the RFLP analysis; arrows indicate the C57 mtDNA haplotype. C57[C57]CD4+ lane displays a band of 664 pb which corresponds to an uncompleted digestion of the PCR product in the RFLP analysis and a high exposure of the image. Representative experiment ($n = 3$). **c** Exosome and mitochondrial transfer to unpulsed or SEE-pulsed Raji B cells (CMAC) from J77 T cells expressing the exosomal protein CD63GFP, the mitochondria-targeted mitoYFP or the mtDNA-binding protein TFAM-DsRED. Dot plots, Cell populations after co-culture. Red boxes enclose Raji B cells that have acquired exosomal or mitochondrial fluorescent markers (percentage from total Raji cells). Graphs: Percentage of Raji B cells acquiring fluorescence upon IS formation from 3 to 5 independent experiments; mean, t-test *P-value < 0.05, **P-value < 0.001, ***P-value < 0.0001. **d** Workflow for mtDNA transfer detection by RFLP during immune cognate interactions. **e** Detection of exogenous C57[C57] mtDNA from donor OT-II CD4+ T cells in recipient C57[NZB] DCs upon antigen stimulation (OVA). Gels, RFLP analysis of the mitochondrial genomes ($n = 2$). Chart: Intensity profile for RFLP. **f** RFLP analysis of the mitochondrial genomes of C57[C57] and C57[NZB] mice. Left gel shows RFLP analysis from C57[C57] T lymphocytes and C57[NZB] DCs. Right gel: lack of exogenous C57[C57] mtDNA from donor DCs in recipient C57[NZB] CD4+ T cells. Chart: intensity profile for RFLP analysis. Representative experiment ($n = 3$)

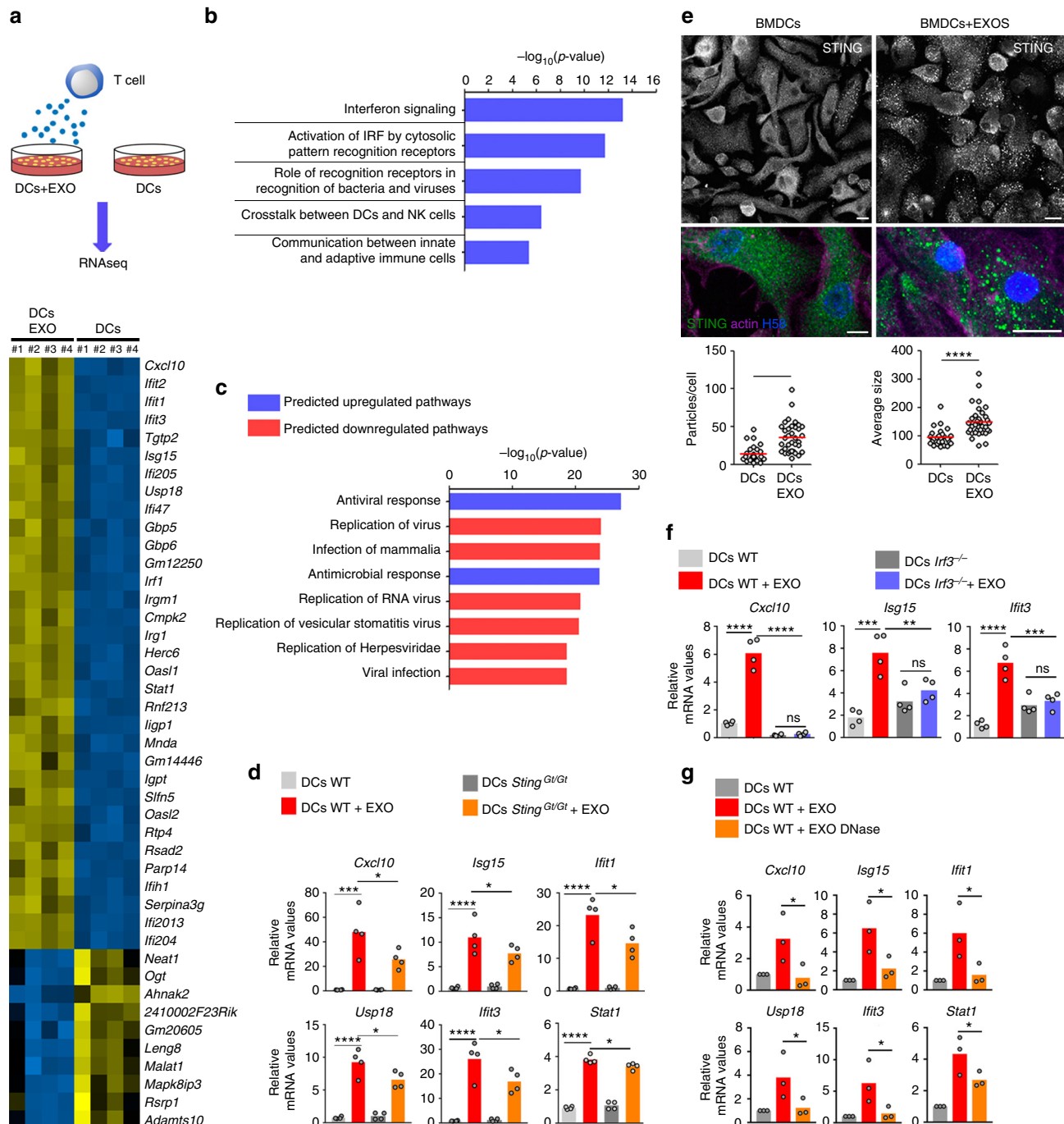

**Fig. 5** Exosomes prime antiviral innate immune responses in DCs. **a** Upper panel: Workflow for RNAseq analysis of DCs upon T cell exosome acquisition. Lower panel: RNAseq heat maps of four DC biological replicates with or without exosome uptake (DCs + EXO vs DCs). The panel shows upregulated (yellow) or downregulated (blue) genes with the highest significant fold changes. **b** GO annotation of the biological processes differently regulated upon exosome addition; p-values are presented for the top-ranking biological processes. **c** Ingenuity analysis predictions of DC pathways upregulated or downregulated upon acquisition of T cell exosomes. **d** Quantitative real-time PCR (qRT-PCR) of antiviral response genes in wild-type DCs and $Sting^{Gt/Gt}$ DCs upon exosome addition. Mean, $n = 4$, t-test ***P-value < 0.0001 and ****P-value < 0.00001. **e** Upper panel: Immunofluorescence microscopy images showing STING aggregation (green) in DCs upon exosome addition. For clarity, DCs were stained for actin with phalloidin (purple) and for nuclei with HOECHST 58 (blue). Bar, 10 μm. Lower panel: Quantification of STING aggregate size and number upon exosome addition. Mean, $n = 26$ (control), 35 (STING), t-test ***P-value < 0.0001 and ****P-value < 0.00001. **f** qRT-PCR analysis of antiviral response genes in wild-type DCs and $Irf3^{-/-}$ DCs upon exosome addition. Data (**d**, **f**) show quantification of mRNA levels of a representative experiment with four mice per genotype: t-test *P-value < 0.05, **P-value < 0.001, ***P-value < 0.0001, and ****P-value < 0.00001. **g** qRT-PCR analysis of antiviral response genes in wild-type DCs upon addition of exosomes left untreated or pre-treated with DNase. Data shows mean from three independent experiments. t-test *P-value < 0.05

DCs upregulated ISGs upon immune cognate interactions with OT-II CD4$^+$T cells (Fig. 6a). Control or SEE-pulsed Raji B cells were co-cultured with CD63GFP-expressing J77 T cells and then sorted according to the level of GFP-labeled exosome uptake (Fig. 6b). Raji B cells with higher acquisition of T cell exosomes showed more potent antiviral responses, as measured by their increased expression of *Cxcl10* (Fig. 6b), supporting that exosome transfer during antigen-driven immune contacts primes antiviral responses in DCs.

To assess the effect of exosome-boosted antiviral responses in recipient cells, we incubated DCs with T cell exosomes, infected them with GFP-expressing recombinant vaccinia virus, and monitored virus spreading and infection. Pre-incubation of DCs with T cell exosomes reduced viral infection, as measured by the number of vaccinia-GFP-positive DCs (Fig. 6c). Antigen-dependent interactions between DCs and CD4$^+$ T cells increased protection against further vaccinia virus infection of the DCs (Fig. 6d and Supplementary Fig. 5a). This effect was dependent on T cell exosomes, since inhibition of their biogenesis by pre-treatment of T cells with increasing concentrations of manumycin A, reduced DCs protection against subsequent viral infection (Fig. 6e), without interfering significantly with T cell viability and activation (Supplementary Fig. 5b, c). Collectively, these results indicate that T cell exosomes delivered during antigen-dependent contacts boost inflammatory responses of DCs, protecting them against subsequent viral infection.

## Discussion

The establishment of immune cognate interactions between T cells and APC is a key element of T cell activation, i.e., the conversion of naive T cells into effector cells, leading to the efficient clearance of pathogens through the adaptive arm of the immune response. This study demonstrates that this signal displays retrograde feedback, in which the T cell imposes additional changes to the activity of the APC, priming it to respond better in the case of subsequent infections by the same pathogen, or a similar one. We have also identified EVs as the vessels of the information that triggers these changes in DCs (the type of APC studied throughout), promoting an antiviral response with induction of IFN-related stimulated genes. A key observation is that these effects require antigenic stimulation. This is crucial because it ensures that the acquisition of inflammatory (or antiviral) traits is limited to situations in which a specific stimulus, i.e., the pathogen, is triggering these responses. It also has the potential to drastically alter the fate of the DC. DCs supposedly die in lymph nodes after cognate interactions. This is based on evidence of the onset of autoimmune disease when DC apoptosis is inhibited[39]. We postulate that, even if DC are eliminated after activating naive T cells in an antigen-specific-dependent manner, T cell exosomes could salvage the DC by reverting the onset of the apoptotic program. This is consistent with the observation that DCs receive pro-survival signals after IS formation, reducing the percentages of apoptosis in vitro and in vivo[7]. Hence, the reported disappearance of DCs from the lymph nodes after the IS[40] could be explained due to their programmed cell death but also due to their migration to other tissues, particularly if they have been fine-tuned by T cell exosomes. In fact, DCs can live for 15 days in the lymph node after an immune challenge, and are able to present antigens ex vivo to CD4$^+$ T cells[41]. There is scant evidence of this type of priming, but some indirect evidence points to the potential importance of this mechanism. For example, tissue-resident memory T cells (T$_{RM}$) have been shown to allow superior protection to homologous infection compared to circulating memory T cells. These T$_{RM}$ can become instructors of innate immune system by initiating a local antiviral state after

reinfection that depends on the secretion of pro-inflammatory cytokines such as TNF, IL-2, or IFN-γ that rapidly initiate an innate immune response in the infected tissues[42–44]. These experimental observations suggest the existence of cross-priming elements that are transferred by the exposure to one pathogen to improve the response against other pathogens. Hence, it is tempting to speculate a role for exosomes and DNA transfer in this enhanced antiviral protection exerted by tissue-resident memory T cells.

The mechanism proposed here would function as a mechanism of innate memory, adapting the DC response during antigenic challenge. Obviously, DCs do not bear genetic mechanisms to modulate their ability to recognize antigens similar to the recombination observed in B cells and T cells. However, increasing evidence has demonstrated that DCs and macrophages bear long-lasting modifications to their epigenetic status after infection or vaccination, leading to enhanced responsiveness upon secondary stimulation by microbial pathogens, increased production of inflammatory mediators, and enhanced capacity to eliminate infection[45]. We propose that T cell EVs are powerful DC reprogramming tools that enhance the adaptation of the innate cell against the pathogen. Although this is obviously a short-ranged mechanism, the delivery of T cell EVs to circulating fluids and their subsequent capture by distant APC could play a significant role in boosting vaccination and the induction of long-distance preparation against infection.

A most striking result is the specificity of the type of cellular response evoked by the DNA packaged into EVs: type I IFN response and expression of ISGs. Although nuclear DNA is the main endogenous ligand for cGAS, growing evidence indicates a role for mtDNA as a ligand for cGAS in certain conditions. Mitochondrial DNA as well as other mitochondrial constituents such as N-formyl peptides (NFPs), mitochondrial lipids such as cardiolipin, or nuclear-encoded mitochondrial proteins, can elicit a local or systemic innate immune response in conditions of massive episodes of cell damage[36]. Intriguingly, mtDNA can also be released in a rapid, "catapult-like" manner by neutrophils to create "extracellular traps" that exert antibacterial activity and activate the release of pro-inflammatory cytokines[46,47]. In this regard, secretion of interferogenic mtDNA by neutrophils in systemic lupus erythematosus (SLE) patients contributes to disease by activating the production of potent IFN type I responses[48,49]. Intracellularly, the accumulation of mtDNA molecules in the cytosol upon mtDNA stress has been also shown to activate type I IFNs response through the cGAS/STING pathway[50], contributing to the acquisition of an antiviral state. Our results demonstrate that in DCs, the DNA present in EVs activate type I IFN responses and expression of ISGs. The contribution of cytoplasmic sensors, e.g., STING, appears to be only partial, and other DNA receptors localized at the endosome or the plasma membrane could also detect the DNA present on the surface or the lumen of EVs and activate interferon response factors, thus contributing to the induction of a potent antiviral response. Then, understanding which sensors or receptors detect the DNA transported in EVs; and whether the mechanisms (endocytosis, phagocytosis, back-fusion of endosomes) by which vesicular content is acquired by recipient cells affect DNA responses are questions that deserve further investigation. In this study we found that the DNA-sensing responses were abrogated in an IRF3-dependent manner. Also, elimination of exosomal DNA, either by DNase treatment or overexpression of DNase II fused to the intra-luminal domain of the exosome-enriched protein CD63, significantly reduced the expression of antiviral-related genes. However, we cannot rule out the contribution of other EVs constituents in triggering innate immune signaling in DCs.

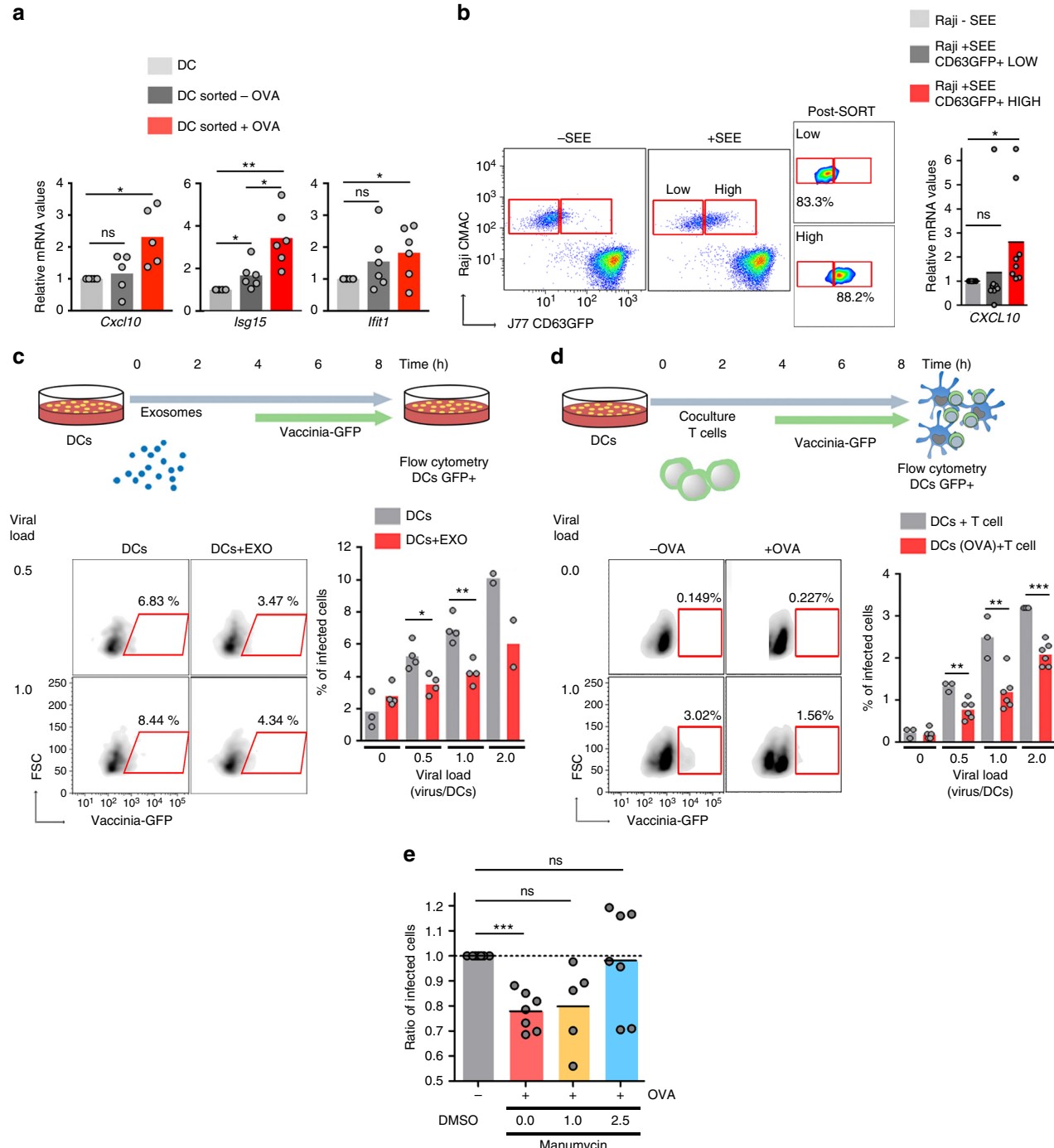

**Fig. 6** Antigen-driven immune interactions prime DCs to pathogen infection. **a** qRT-PCR analysis of the DC antiviral response upon antigen-dependent contacts. Gene expression was analyzed in sorted DCs after 16 h conjugate formation. Data shows mean from five independent experiments. t-test *P-value < 0.05. **b** Left: Flow cytometry analysis of exosome uptake after co-culture of J77 T cells stably expressing CD63GFP with unprimed or SEE-primed Raji B cells. After immune interactions, Raji B cells (CMAC) were sorted according to the level of exosome uptake (CD63GFP content). Dot plots show the GFP signal in pre-sort and post-sort Raji populations. Right: Chart shows qRT-PCR analysis of *Cxcl10* in low and high exosome uptake SEE-primed Raji B cell populations. Mean; n = 3. t-test *P-value < 0.05. **c** Upper panel: Time flow for the analysis of exosome-mediated antiviral protection. DCs were incubated for 4 h with T cell exosomes, infected with vaccinia-GFP, and analyzed for the level of infection by flow cytometry. Lower left: Dot plots of DCs infected with vaccinia-GFP with or without exosome pre-treatment. Lower right: Percentage of infected DCs at different viral loads with or without exosome pre-treatment. Mean; n = 4; t-test *P-value < 0.05, **P-value < 0.01. **d** Upper panel: Time flow for the analysis of exosome-mediated antiviral protection after immune cognate interactions. Unloaded or OVA peptide-loaded DCs were incubated for 4 h with CD4[+]OT-II T cells, infected with vaccinia-GFP, and analyzed for the level of infection by flow cytometry. Lower left: Dot plots of DCs (gated on CD11c) infected with vaccinia-GFP in the presence or absence of OVA peptide. Lower right: Percentage of infected DCs at different viral loads in the presence or absence of antigen-dependent immune contacts. Mean; n = 3–6; t-test *P-value < 0.05, **P-value < 0.01, ***P-value < 0.0001. **e** Unloaded or OVA peptide-loaded DCs were incubated for 4 h with CD4[+]OT-II T cells pre-treated with increasing amounts of manumycin, infected with vaccinia-GFP, and analyzed for the level of infection by flow cytometry. Graph shows the ratio of infected DCs. Data (**b**, **c**, **d**, **e**) are Mean from quantification of five to seven independent experiments: t-test *P-value < 0.05, **P-value < 0.001, and ***P-value < 0.0001

The molecular mechanism of inhibition of exosome production by manumycin A in T cells was initially related to neutral sphingomyelinase inhibition and lack of ceramide production, as reported previously[9]. Our data show that manumycin A prevents exosome production in T cells without interfering with activation at 1.0 and 2.5 μM. In addition, manumycin effect on exosome biogenesis has been recently reported to be dependent on its inhibition of Ras farnesyl transferase[51]. Therefore, the effect of manumycin on T cells maybe mediated by a combination of effects on these systems, leading to a final defect on exosome production. Manumycin has also been described to induce a certain degree of cell death mainly in tumor cells through Ras Farnesyl transferase and IKKbeta inhibition activity[51,52]. Although our data show a decrease of 13% in T cell viability at manumycin 2.5 μM, since the ratio of T cells: DCs in the activation assay is 5:1, the viable, active T cells present would constitute a 4.35:1 effective ratio, enough to promote the antiviral response in DCs. Such a modest degree of cell death would not explain the complete loss of antiviral protection by manumycin.

Our observations of mtDNA and mtDNA-binding proteins loading into EVs in an autophagy-independent manner indicate the existence of an additional targeting mechanism. The role of mitochondrial component release as a danger indicator to surrounding cells is well documented[36], but our our data suggest that the MVB endosomal route can target these components actively into exosomes. Although the signals that mediate the loading and incorporation of mitochondrial components into late endosomes/ MVBs remain unidentified, an attractive possibility is that small organelle fragments containing mitochondrial nucleoids might bud toward the cytosol depending on their oxidative status, and are subsequently loaded and incorporated into the endolysosomal compartment for their degradation or secretion. Cytosolic accumulation of mtDNA has been described following mitochondrial dysfunction and inflammatory stimulation[36,53], and a similar process may operate in the release of mtDNA to the cytoplasm and its subsequent incorporation into ILVs during MVB maturation. Alternatively, mitochondria-derived vesicles (MDVs) transport mitochondrial components into lysosomes, peroxisomes, and MVBs[54], and they might participate in the loading of mitochondrial components into exosomes. Our results demonstrating the presence of oxidized DNA in exosomes suggest that these EVs may cooperate to manage cellular damage by releasing mitochondrial proteins and oxidized DNA, acting as a "escape valve" to partially expel mitochondrial damage outside the cell. Recently, a role for exosomes in the secretion of harmful genomic DNA has been documented[55]. Supporting this role of exosomes in the preservation of cellular homeostasis, we found that inhibition of exosome biogenesis and secretion impairs mitochondrial structure and function. A recent study identified a role for the secretion of large vesicles containing protein aggregates and organelles in *Caenorhabditis elegans* in maintaining mitochondrial homeostasis[56]. Whereas exosomes facilitate the secretion of harmful or damaged mitochondrial components and sustain mitochondrial function is a possibility that warrants further investigation.

Transfer of entire mitochondria between cells or acquisition of mitochondria from the environment in a functional and transmissible way has been demonstrated in culture cells[57–59], and in vivo in a cancer context[60–62]. Recent reports suggested that the intercellular transmission in vivo of mitochondria and/or mitochondrial components exerts a beneficial effect on the recipient damaged cells[63–66].The benefits in all cases are attributed to the assumption that entire healthy functional mitochondria travel and colonize the recipient cells; however, the use of interspecies mix for donor cells in some cases and the lack of evidence of long-lasting colonization of the recipient cells by the foreign

mtDNA question this interpretation. The role of the intercellular transmission of defined mitochondrial components and specifically mtDNA as part of a complex intercellular signaling mechanism, raises the possibility that the observed beneficial effects of transmission of mitochondrial components would be consequences of activation of survival pathways rather than the transmission of fully functional organelles able to colonize the recipient cells. This could, for instance, explain the positive effects of interspecies (human/mice) experiments, since it is well described the interspecies incompatibility between nuclear and mtDNA.

In summary, this study provides a physiological mechanism for the transfer of mitochondrial components and genome transfer in the context of immune cognate interactions, laying the foundation of a model in which the transfer of DNA between immune cells constitutes a signal-dependent mechanism for alerting responses in innate immune cells.

## Methods

**Antibodies and reagents**. The following antibodies were used for Western blotting: mouse anti-human CD63 (Calbiochem, OP171; 1:1000); mouse 5A6 anti-human CD81 (Santa Cruz, sc-23962; 1:500); mouse anti-human TSG101 (Abcam, ab83; 1:1000); mouse anti-GFP (Clontech, 632381; 1:2000); anti-TFAM (Abcam, ab47517; 1:500), anti-HRS (Abcam, ab72053; 1:500), anti-COX1 (Human Complex IV subunit I, Invitrogen, 459600; 1:1000), anti-VDAC1 (Abcam, ab14734; 1:1000), and anti-Cyt C (Abcam, ab110325; 1:1000), anti-LC3 (Cell Signaling, 2775 s; 1:1000); anti-MnSOD (Enzo Life Sciences, ADI-SOD-110; 1:1000), anti-ATAD3 (Abnova, H00055210-D01), anti-Tubulin (Sigma, T6199: 1:2000), anti-nSMase2 (RD System, MAB7184;1:1000) and anti-Rab27a (RD System, AF7245; 1:1000). Peroxidase secondary antibodies were from Thermo Scientific: Goat anti-mouse peroxidase (31430; 1:5000) and goat anti-rabbit peroxidase (31460; 1:5000).

For immunofluorescence and flow cytometry the following antibodies were used: anti-CD63 (clone Tea 3/18, generated in the laboratory; 1 μg/ml), anti-HRS (Abcam, ab72053; 1:200), anti-EE1A (Santa Cruz, sc-137130; 1:200), anti-LBPA (Tebu-Bio; ML062915-21), anti-ceramide (Sigma, C8104-50TST: 1:200), anti-8 hydroxyguanosine (Abcam, ab62623; 1:500), anti-DNA AC-30-10 (Progen, 61014; 1:50), anti-Tmem173/Sting antibody (Proteintech, 19851-1-AP; 1:100) and anti-SSBP1 (Novus Bio, NBP1-80720: 1:200), and anti-CD69 (BD Pharmigen, 552879; 1:200)

**Human cells**. Human T lymphoblasts were isolated from buffy coats obtained from healthy donors by separation on a Biocoll gradient (Biochrom). After 30 min of adhesion at 37 °C, non-adherent cells were cultured in RPMI (Gibco) for 2 days in the presence of 5 μg/ml phytohemagglutinin (PHA) to induce lymphocyte proliferation. T lymphoblasts were maintained by addition of IL-2 (50 U/ml) to the medium every 2 days over 8–10 days. For T lymphoblast restimulation, cells were incubated overnight with 50 ng/ml PMA and 750 ng/ml ionomycin.

Human peripheral blood mononuclear cells (PBMC) were isolated from buffy coats obtained from healthy donors by separation on a Lymphoprep gradient (Nycomed, Oslo, Norway) according to standard procedures. Buffy coats of healthy donors were received from the Blood Transfusion Center of Comunidad de Madrid, and all donors signed their consent for the use of samples for research purposes. All the procedures using primary human cells were approved by the Ethics Committee of the Hospital Universitario de la Princesa. Monocytes were purified from PBMC by a 30 min adherence step at 37 °C in RPMI supplemented with 10% fetal calf serum. Non-adherent cells were washed off and the adhered monocytes were cultured in RPMI, 10% FCS containing IL-4 (10 ng/ml, R&D Systems Inc, Minneapolis, MN USA) and GM-CSF (200 ng/ml, Schering-Plough, Madrid, Spain). Cells were cultured for 6 days, with cytokine re-addition every other day, to obtain a population of immature hDC.

The human Jurkat-derived T cell line J77 E61 (TCR Vα1. 2 Vβ8) (from NIH AIDS reagent program) and the lymphoblastoid B cell line Raji (Burkitt lymphoma) (from ATCC, CCL-86™) were cultured in RPMI (Gibco) containing 10% FBS. Stable cell line clones overexpressing CD63-GFP were generated by transfection and selection with G418 (1 mg/ml, Sigma).

HEK293T cells (from ATCC, CRL-3216™), were cultured in DMEM (Gibco) containing 10% FBS. For co-localization experiments HEK293T cells were treated with 1 μM FCCP 4 h (low dose), then washed and immediately fixed with PFA 6%. HEK923T cells were used as a bona-fide cell factory for exogenous protein synthesis and organelle compartmentalization.

Cells were regularly tested for mycoplasma contamination by PCR.

**Mice and cell differentiation and activation**. Mice were housed under specific pathogen-free conditions at the Centro Nacional de Investigaciones Cardiovasculares Carlos III (CNIC), and experiments were approved by the CNIC Ethical

Committee for Animal Welfare and by the Spanish Ministry of Agriculture, Food, and the Environment. Animal care and animal procedures license were reviewed and approved by the local Ethics Committee for Basic research at the CNIC Ethical Committee for Animal Welfare and the Organo Encargado del Bienestar Animal (OEBA) del Gabinete Veterinario de la Universidad Autonoma de Madrid (UAM). C57BL/6 JOlaHsd (or C57BL/6), C57BL/6 OT-II, C57BL/6JOlaHsd with NZB mtDNA (or C57BL/6NZB) and C57BL/6NZB OT-II mice were bred under specific pathogen-free conditions at the CNIC (Madrid, Spain) in accordance with European Union recommendations. The C57BL/6JOlaHsd strain does not harbor the nicotinamide nucleotide transhydrogenase (NNT) spontaneous mutation that renders the encoded enzyme undetectable, and which is characteristic of the C57BL/6J strain provided directly by Jackson Laboratories. C57BL/6NZB OT-II mice were obtained by backcrossing C57BL/6 OT-II transgenic males with C57BL/6NZB females. $Sting^{Gt/Gt}$ (STING-deficient) mice were provided by Dr. Gloria González-Aseguinolaza, Gene Therapy and Regulation of Gene Expression Program, Center for Applied Medical Research, Health Research Institute of Navarra (IdisNA), Pamplona, Spain. These mice have a single nucleotide variant (T596A) which is a null mutation resulting in failure to produce STING protein[67]. $Irf3^{-/-}$ mice were kindly provided by Prof. Adolfo García Sastre, Icahn School of Medicine at Mount Sinai, New York, USA.

CD4[+] T cells were obtained from spleen-cell suspensions obtained from C57BL/6 OT-II transgenic mice. Splenic cells were incubated for 30 min at 4 °C with a biotinylated anti-CD4 antibody (BD Pharmingen) followed by extensive washing with PBS, 1% BSA, 5 mM EDTA. Cell suspensions were then incubated for 20 min at 4 °C with streptavidin microbeads (MACS; Miltenyi Biotec) and washed with PBS, 1% BSA, 5 mM EDTA. CD4[+] T cells were obtained by positive selection using the auto-MACS Pro Separator (Miltenyi Biotec).

Bone marrow-derived dendritic cells (BMDCs) were obtained either from C57BL/6, C57BL/6NZB, $Sting^{Gt/Gt}$, or $Irf3^{-/-}$ mice by incubating primary bone marrow cells for 9–10 days in RPMI (Gibco) supplemented with 1% sodium pyruvate, 50 U/ml penicillin, 50 µg/ml streptomycin, $5 \times 10^{-5}$ M 2-mercaptoethanol, and 10% FBS (all from Invitrogen) supplemented with 20 ng/ml recombinant GM-CSF (ImmunoTools).

For T cell blasting, naive CD4[+] T cells were isolated from spleen and lymph nodes and cultured for 48 h with 2 µg/ml concanavalin A (Sigma). Cells were then incubated with 50 U/ml human recombinant IL-2 (Glaxo) every 2 days for at least 7 days to obtain differentiated T lymphoblasts. For antigen-specific restimulation, T lymphoblasts were conjugated with BMDCs and stimulated overnight in vitro with 100 ng/ml lipopolysaccharide (LPS) in the presence or absence of 10 µg/ml ovalbumin (OVA) peptide.

**Cell transfection and plasmids and interference RNA**. J77 cells were transfected with the corresponding plasmids by electroporation; $20 \times 10^6$ cells were resuspended in Opti-MEM (Gibco; $5 \times 10^7$ cells per ml) with 20 µg of plasmid DNA and electroporated with a Gene Pulser Xcell electroporator (Bio-Rad) at 1200 µFa, 240 mV, 30 ms in 4 mm Bio-Rad cuvettes (Bio-Rad). Fluorescence-positive cells were FACS sorted, cloned, and cultured in RPMI containing 2 mg/ml G418 (Invitrogen). HEK293T cells were transfected using Lipofectamine 2000 (Invitrogen) according to the manufacturer's instructions.

The following plasmids were used: mitoDsRed and mitoYFP, (kindly provided by Prof. L. Scorrano, Venetian Institute for Molecular Medicine, Italy), TFAM-GFP and TFAM-DsRED (generated by inserting the human TFAM cDNA into the pEGFP and DsRED plasmids), EGFP-Rab7a Q67L, LC3-GFP and ATG5-GFP (obtained from Addgene: plasmids 28049, 24920 and 22952, respectively), shRNA pLKO control plasmid, shnSMase2 and shRab27a shRNAs (obtained from Open Biosystems), and CD63-GFP[9].

Construct of DNase II with CD63 intraluminal domain: Following Gibson Assembly Protocol (New England Biolabs) hDNaseII (Human DNase II Gene ORF cDNA clone in cloning vector, HG16372-G, Sino Biological Inc) was inserted in CD63-GFP plasmid in the intraluminal domain. CD63-GFP plasmid was cut with EcoRI. The following primers were used to amplify insert: Fw_DNaseII-EcoRI: CGAGCTCAAGCTTCGATGATCCCGCTGCTGCT Rv_DNaseII-EcoRI: TACCGTCGACTGCAGGGATCTTATAAGCTCTGCTGG GC

T cell lines and human T lymphoblasts were transfected with a specific double-stranded siRNA targeting human LC3 (GGUGCCUGUAGGGUGAUCAA) or negative control siRNA (Eurogentec), using the Gene Pulser II electroporation system (Bio-Rad Laboratories) or the Nucleofector system (Amaxa Biosystems). When indicated, at 24 h post transfection, cells were cultured in exosome-depleted medium for 20 h and exosomes were isolated from supernatants.

**Lentiviral infection**. J77 cells with downregulated nSMase2 and Rab27a expression were generated by infection with lentiviral vectors expressing specific shRNAs. HEK293T cells were co-transfected using the Lipofectamine 2000 system (Invitrogen) with pCMV-ΔR8.91-(Delta 8.9), pMD2.G-VSV-G, and plasmids encoding shRNAs targeting nSMase2, Rab27a, or the corresponding shRNA pLKO control plasmids (Open Biosystems). Supernatants were collected after 48–72 h, filtered (0.45 µm), and added to J77 or HEK293 cell cultures. Cells were centrifuged ($2000 \times g$, 2 h) and incubated for 4 h at 37 °C. Medium was replaced with RPMI containing puromycin (4 µg/ml).

**Conjugate formation and assessment of mtDNA transfer**. To assess mtDNA transfer during immune cognate interactions, differentiated C57BL/6NZB receptor BMDCs were loaded with calcein (37 °C, 20 min), extensively washed, and then co-cultured in the presence of ovalbumin (OVA) peptide (10 µg/ml) with C57BL/6 CD4[+] OT-II T cells (5:1 T cell/DC ratio; 16 h). To assess mtDNA transfer from BMDCs to CD4[+] OT-II cells, C57BL/6 BMDCs and C57BL/6NZB OT-II CD4[+] T cells were conjugated (1:5 T cell/DC ratio; 16 h). Conjugates were blocked with Fc-block (CD16/CD32, BD Pharmingen), washed in PBS, 1% BSA, and incubated with antibodies against MHCII and CD3 (BD Pharmingen) diluted in PBS, 1% BSA. Viable cells were identified by propidium iodide exclusion. Singlet cells were discerned with a stringent multiparametric gating strategy based on FSC and SSC (pulse width and height). CD4[+] T cells and BMDCs were distinguished by calcein staining and MHCII and CD3 fluorescence, and sorted on a FACSAria flow cytometer (BD Biosciences). DNA was extracted, and C57BL/6 and NZB mtDNAs were detected by restriction fragment length polymorphism (RFLP) analysis as follows. DNA (50 ng) was PCR-amplified using the RED Extract-N-Amp PCR Ready Mix Kit (Sigma Aldrich) with the oligonucleotides 5' AAGCTATCGGG CCCATACCCCG 3' (Fw) and 5' GTTGAGTAGAGTGAGGGATGGG 3' (Rv) at a concentration of 250 nM (Tm: 58°). After PCR amplification, samples were digested with BamH1 (10 units) for 2 h. Digested samples were loaded on a 2% agarose gel. C57NZB haplotype includes a BamHI restriction site; enzyme digestion results in two smaller fragments of 414 and 250 pb. Digestion of PCR products from C57C57 haplotype results in a single band of 664 pb corresponding to the full length PCR product.

**Conjugate formation of cell lines and mitoDsRed transfer**. To distinguish Raji B cells from T cells, Raji cells were loaded with the cell tracer CMAC (7-amino-4-chloromethylcoumarin, Invitrogen). For antigen stimulation and IS formation, Raji cells were pulsed for 30 min with 0.5–1 µg/ml of Staphylococcus enterotoxin superantigen (SEE, Toxin Technology) and mixed with Jurkat cells expressing CD63-GFP, TFAM-dsRED, or mitoYFP at a ratio of 1:5 for 16–24 h at 37 °C. Cell conjugates were analyzed by FACS in a FACS LSRFortessa Cell Analyzer (BD Biosciences) and data were analyzed with FACSDiva software (BD Biosciences).

To assess gene expression after IS formation, stable J77 CD63-GFP cells and CMAC-stained Raji cells were conjugated as described. After 16 h, cells were sorted according to CMAC staining and CD63 green fluorescence. The Raji population was split into two groups according to CD63-GFP uptake: Raji GFP + Low (cells acquiring low amounts of CD63) and Raji GFP + High (acquiring high amounts of CD63). After sorting, total RNA was isolated and qPCR performed to analyze antiviral response genes.

**Exosome purification and sucrose-gradient purification**. Cells were cultured in medium supplemented with 10% exosome-depleted fetal bovine serum (FBS, Invitrogen; FBS was depleted of bovine exosomes by overnight centrifugation at $100,000 \times g$ for at least 16–20 h. Exosomes were obtained by serial centrifugation as described[20]. Briefly, cells were pelleted ($300 \times g$ for 10 min) and the supernatant was centrifuged at $2000 \times g$ for 20 min to remove debris and dead cells. The collected supernatant was then ultracentrifuged at $10,000 \times g$ for 40 min at 4 °C (Beckman Coulter Optima L-100 XP, Beckman Coulter) to remove debris, apoptotic bodies, and shedding vesicles. Exosomes were pelleted by ultracentrifugation at $100,000 \times g$ for 70 min at 4 °C, washed in PBS, and collected by ultracentrifugation at $100,000 \times g$ for 70 min. DNase treatment was performed with RQ1 RNase-Free DNase (Promega) according to the manufacturer's instructions.

Exosomes were isolated by sucrose density gradient. Briefly, exosomes were obtained by serial centrifugation of 200 ml cell culture supernatant. Isolated exosomes were layered on top of a discontinuous sucrose gradient, ranging from 2.5 M sucrose at the bottom to 0.4 M sucrose at the top, in an SW40 centrifuge tube (14 × 95 mm; Beckman Coulter). Tubes were centrifuged overnight (18–20 h) in a SW40 rotor at $192,000 \times g$ at 4 °C. One ml fractions were collected from the top of the gradient and were resuspended in an equal volume of acetone for protein precipitation and Western blot analysis or were treated as indicated for DNA isolation and PCR analysis.

**Exosome adsorption to aldehyde sulfate beads and detection**. Exosomes were obtained as described above, resuspended in PBS, and coupled to 4 µm aldehyde sulfate beads (Invitrogen) 15 min in rotation at room temperature. Then, coupled beads where incubated in overnight rotation in PBS containing 0.1% BSA and 0.01% sodium azide. Beads were washed twice with PBS, 0.1% BSA, 0,01% sodium azide. For surface staining, beads were washed and incubated with the indicated primary antibodies for 30–45 min on ice. For intraluminal staining, beads were fixed and permeabilized in PBS containing 1% Paraformaldehyde and 0.01% Triton X-100 for 20 min on ice, washed, and incubated with primary antibodies for 1 h on ice. Beads were then washed and incubated with the appropriate secondary antibodies for 30 min on ice. Beads were analyzed with a FACS LSRFortessa Cell Analyzer (BD Biosciences) and data were analyzed with FACSDiva software (BD Biosciences). Negative controls were obtained with permeabilized and non-permeabilized exosome-coupled beads incubated with the indicated secondary antibodies.

**Nanoparticle tracking analysis**. Exosome numbers and size distribution were measured from the rate of Brownian motion in a NanoSight LM10 system, which is equipped with fast video capture and particle-tracking software (NanoSight, Amesbury, UK). Briefly, 0.5 ml of diluted exosome fraction was loaded into the sample chamber of an LM10 unit (Nanosight, Amesbury, UK) and three 30-s videos were recorded of each sample. Data were analyzed with NTA 2.1 software (Nanosight). Samples were analyzed using manual shutter and gain adjustments, which resulted in shutter speeds of 15 or 30 ms, with camera gains between 280 and 560. The detection threshold was kept above 4; blur: auto; minimum expected particle size: 50 nm. Samples were diluted before analysis to concentrations between $2 \times 10^8$ and $20 \times 10^8$ particles/ml.

**Deep-sequencing analysis**. Total RNA was extracted with the RNeasy kit (Qiagen). Contaminating genomic DNA was removed using the specific column from the kit. RNA integrity and quantity were determined in an Agilent 2100 Bioanalyzer. Equal RNA amounts from four individual samples per group were analyzed in the Illumina Genome Analyzer IIx. Sequencing reads were pre-processed by means of a pipeline that used FastQC to assess read quality and Cutadapt to trim sequencing reads, eliminate Illumina adapter remains, and discard reads shorter than 30 bp. The resulting reads were mapped against the mouse transcriptome (GRCm38, release 76; aug2014 archive) and quantified using RSEM v1.17. Data were then processed with a pipeline that used the EdgeR Bioconductor package for normalization and differential expression analysis in a paired strategy.

The number of reads obtained per sample was in the range of 12 to 17 million. The percentage of reads eliminated in the pre-processing step was below 1%. Genes with at least 1 count per million in at least four samples (11,383 genes) were considered for further analysis. Fold change and log(ratio) values were calculated to represent gene expression differences between conditions. Sets of genes differentially expressed across conditions were analyzed for functional associations using the Ingenuity Knowledge Database (IPA, http://www.ingenuity.com).

**Exosome proteomic analysis**. Exosomes from mouse T lymphoblasts were isolated by serial ultracentrifugation, and protein extracts (200 µg) were digested using the FASP method as previously described[68]. After digestion, the resulting peptides were desalted onto C18 OMIX tips (Agilent Technologies). Peptides were separated into four fractions by cation exchange chromatography with MCX cartridges (Waters) using graded concentrations of ammonium formate, pH 3.0 in acetonitrile (ACN), and desalted again before analysis by liquid chromatography–tandem mass spectrometry (LC-MS/MS).

Analyses were performed using a nano-HPLC Easy nLC 1000 chromatograph coupled to a Q-Exactive hybrid quadrupole orbitrap mass spectrometer (Thermo Scientific). Peptides were loaded in an analytical C18 nano-column (EASY-Spray column PepMap RSLC C18, 75 µm internal diameter, 3 µm particle size, and 50 cm length, Thermo Scientific) and separated in a continuous gradient consisting of 8–30% B for 60 min and 30–90% B for 2 min (B = 90% ACN, 0.1% formic acid) at 200 nl/min. The chromatographic run acquired an FT-resolution spectrum of 140,000 ions in the mass range of $m/z$ 390–1600 followed by data-dependent MS/MS spectra of the 15 most intense parent ions. Normalized HCD collision energy was set to 28% and the parent ion mass isolation width was 2-Da.

Peptides were identified from MS/MS spectra using the SEQUEST HT algorithm integrated in Proteome Discoverer 2.1 (Thermo Scientific). MS/MS scans were matched against a mouse protein database (UniProt 2016_07 Release) and the parameters were selected as follows: a maximum of two missed trypsin cleavages, a precursor mass tolerance of 800 ppm, and a fragment mass tolerance of 0.02 ppm. Cysteine carbamidomethylation was chosen as the fixed modification and methionine oxidation as the dynamic modification. The same MS/MS spectra collections were searched against inverted databases constructed from the same target databases. SEQUEST results were analyzed by the probability ratio method. False discovery rate (FDR) was calculated for peptides identified in the inverted database search results using the refined method[69].

**Detection of oxidized DNA by chromatin immunoprecipitation**. T lymphoblast exosomes were purified by differential ultracentrifugation and lysed in buffer containing 25 mM Tris-HCl, 150 mM NaCl, 2 mM MgCl, 0.5%, NP40, 5 mM DTT. Protein G-coupled magnetic beads (Protein G Dynabeads, Invitrogen) were washed and incubated for 2–3 h with 5 µg anti-8 hydroxyguanosine (8-OHdG) antibody [15A3] (Abcam), 5 µg of anti-DNA AC-30-10 (Progen), or with 5 µg of the isotype control (Santa Cruz, sc-2015). Lysates were precleared with Protein G Dynabeads and then incubated overnight at 4 °C with antibody-coupled beads. Samples were washed five times with lysis buffer and eluted with 50 mM Tris-HCl pH 8.0, 10 mM EDTA, and 1% SDS for 1 h at 65 °C. DNA was purified with the ChIP DNA Clean & Concentration kit (D5205-Zymo Research). qPCR was performed on whole samples with technical duplicates.

Mitochondrial enriched fraction was obtained from T lymphoblasts by resuspending them in mitochondrial extraction buffer (1 mM EDTA pH 8.0, 1 mM DTT, 10 mM NaCl, 3 mM MgCl2, 5 mM Tris-HCl pH 8.0) during 15 min at 4 °C (Osmotic Lysis). Then, lysates were subjected to three freeze/thaw cycles with 5 min liquid nitrogen exposition and 10 min immersion in a water bath at 37 °C. Following lysis the tubes were centrifuged at $500 \times g$ for 10 min at 4 °C.

Supernatants were collected and spun down at $12,000 \times g$ for 20 min at 4 °C. The final pellets (mitochondria enriched fraction) were resuspended in lysis buffer and treated as exosomal fractions.

**Analysis of antiviral response induction by exosomes**. BMDCs were cultured and differentiated in vitro. After culture for 9–11 days, BMDCs were plated in p12 plates at $3 \times 10^5$ per well. Exosomes from T lymphoblasts were isolated by differential ultracentrifugation and 10–15 µg added to BMDCs for 6 h. BMDCs were then washed and frozen for RNA extraction.

**Vaccinia infection**. The recombinant vaccinia virus VACV-GFP was a gift from J. Yewdell (National Institutes of Health, Bethesda, MD). For analysis of exosome-mediated viral protection, BMDCs were cultured as previously described. After 9–11 days, cells were plated in p96 plates at $1 \times 10^5$ per well. T lymphoblast exosomes were purified by differential ultracentrifugation and 5–10 µg were added to BMDCs for 4–6 h. Vaccinia-GFP was added to cells at the indicated BMDC:virus ratios. After incubation for 4 h, cells were harvested, fixed for 10 min with BD Cytofix/Cytoperm (BD Biosciences), and analyzed by flow cytometry. Viability was assessed with the LIVE/DEAD® Fixable Aqua Dead Cell Stain Kit (Molecular Probes).

For the analysis of antiviral protection during IS formation, BMDCs were cultured for 9–11 days and then plated in p24 plates at $2 \times 10^5$ per well. Receptor BMDCs were primed with 100 ng/ml LPS overnight, extensively washed, and co-cultured in the presence of ovalbumin (OVA) peptide (5 µg/ml) with C57BL/6 CD4+ OT-II T cells (5:1 T cell/DC ratio). After conjugate formation for 4 h, vaccinia-GFP was added at a 1:1 ratio (total number of cells in the well) for 4 h. Cells were harvested, blocked with Fc-block (CD16/CD32, BD Pharmingen), and stained with CD4 PE-Cy7 and CD11c PE. Viability was assessed with the LIVE/DEAD® Fixable Aqua Dead Cell Stain Kit (Molecular Probes). Cells were fixed for 10 min with BD Cytofix/Cytoperm (BD Biosciences) and analyzed by flow cytometry.

**Cell viability assay**. In order to measure cell viability upon manumycin treatment and activation, CD4+ OT-II T lymphoblasts were incubated for 24 h with increasing amounts of manumycin (0, 1, and 2.5 µM) and with anti-CD3 (10 µg/ml) and anti-CD28 (2 µg/ml). Cells were harvested and stained with anti-CD4 and DAPI to measure the percentage of DAPI negative (alive) CD4+ cells by flow cytometry.

To ascertain cell viability in the context of immune cognate interactions, CD4 + OT-II T lymphoblasts were pre-treated for 2 h with increasing amounts of manumycin (0, 1, and 2.5 µM), then CD4 T cells were conjugated with OVA-pulsed dendritic cells (5:1 T cell/DC ratio; 4 h). Cells were harvested and stained with anti-CD4 and DAPI to measure the percentage of DAPI negative (alive) CD4 + cells by flow cytometry.

**Flow cytometry**. For cytometry, adherent cells were harvested with 0.25% Trypsin-EDTA (Gibco), blocked with Fc-block (CD16/CD32, BD Biosciences), washed in PBS, 1% BSA, and incubated with primary antibodies diluted in PBS, 1% BSA. Viable cells were identified by HOECHST 58 exclusion. Singlet cells were discerned based on FSC and SSC (pulse width and height). For surface labeling, cells were stained with antibodies diluted in PBS, 0.5% BSA on ice.

For intracellular and surface staining, cells were fixed with 1% formaldehyde and permeabilized with 0.01% Triton X-100 (Sigma). After washing, cells were blocked with Fc-block (CD16/CD32, BD Biosciences), washed in PBS, 1% BSA, and incubated sequentially with primary and secondary antibodies (diluted 1:500) in PBS, 1% BSA. Cells were then analyzed by FACS in a FACS LSRFortessa Cell Analyzer (BD Biosciences). Data were analyzed with FACSDiva software (BD Biosciences).

To detect intracellular ROS, HEK293T cells were incubated with 2′,7′–dichlorofluorescin diacetate (DCDFA, Thermofisher), and to detect mitochondrial mass cells were loaded with mitotracker green (Invitrogen). Cells were analyzed in a FACS LSRFortessa Cell Analyzer (BD Biosciences) and data were analyzed with FACSDiva software (BD Biosciences).

**Western blot**. Cells or exosomes were lysed in RIPA buffer (50 mM Tris-HCl pH 8.0, 150 mM NaCl, 1% Triton X-100, 0.1% sodium deoxycholate, and 0.1% SDS) supplemented with a protease inhibitor cocktail (Complete, Roche). Proteins were separated on 10–12% acrylamide/bisacrylamide gels and transferred to nitro-cellulose or methanol-pre-treated PVDF membranes. Membranes were incubated with primary antibodies (1:1000) and peroxidase-conjugated secondary antibodies (1:5000), and proteins were visualized with LAS-3000.

**Immunofluorescense**. For immunofluorescence assays, cells were plated onto slides coated with fibronectin (50 µg/ml), incubated overnight, and fixed with 2% paraformaldehyde in PBS. Cells were permeabilized with PBS, 0.1% Triton X-100 (Sigma), washed and stained with the corresponding primary antibodies (1:100–1:200) followed by secondary antibodies (1:500) (Life Technologies). When indicated, cells were previously transfected with the indicated plasmids. Cells were

fixed and stained with the indicated antibodies at 24 h post transfection. Samples were examined with a Leica SP5 confocal microscope fitted with a ×63 objective, and images were processed and assembled using Leica software. Quantification of images was done with Fiji software (ImageJ) using Manders' coefficient M2 as a marker for co-localization of mitochondrial markers (SSBP1) with endosomal markers (EEA1 and CD63)[70]. More than one hundred cells were quantified for experiment.

**TIRFm analysis of isolated exosomal particles.** Isolated exosomal particles ($8 \times 10^6$ particles per coverslip) generated as described and quantified with Nanosight were spun in a sucrose cushion (10% in HBSS; $10,000 \times g$, 15 min, 25 °C. HS-4 rotor, Sorvall) over 13 mm diameter precision coverslips (thickness No. 1.5 H; 0.170 mm ± 0.005 mm, 0117530 Marienfeld) coated with 250 µg/ml Poly-L-Lys (P6407 Sigma Aldrich) and fixed in 1% paraformaldehyde (HBSS, 5% sucrose; RT15710 Electron Microscopy Sciences) for 5 min, R/T. Coverslips were treated with Glycine 200 mM in Tris-HCL (pH 7.4) for 10 min, R/T and blocked (3% BSA, 20 µg/ml human gamma-globulins in HBSS) for 1 h R/T. Staining with anti-CD81 (rat anti-mouse clone MT81; 5 µg/ml, 4 °C, o/n) was followed by highly cross-adsorbed Alexa-568-conjugated goat anti-rat Antibody (A-11077 Thermofisher Scientific 1:1000). Samples were then permeabilized with a 0.2% TX-100 solution (HBSS supplemented with 0.5% paraformaldehyde and 1.5% BSA), treated for autofluorescence and stained with anti-DNA (0.001 µg/ml) or anti-TSG101 (20 µg/ml, ab83 Abcam) mouse monoclonal antibodies for 3 h at 4 °C, followed by highly Cross-absorbed Alexa-488-conjugated goat anti-mouse Antibody (A-11029 Thermofisher Scientific 1:1000). Samples were mounted on Prolong Gold antifade mountant (P36930 Thermofisher Scientific). Coverslips were extensively washed with HBSS and a final wash with deionized water. Antibodies were diluted in 1.5% BSA in HBSS. Secondary antibodies were incubated for 1 h at R/T. All the solutions used for coverslip preparation and immunofluorescence were previously ultra-centrifuged at $100,000 \times g$ for 70 min (4 °C) in a swinging bucket rotor (Beckman Coulter) or for 30 min at $13,200 \times g$ in a fixed-angle rotor (Eppendorf). Samples were imaged under a Leica AM TIRF MC M system mounted on a Leica DMI 6000B microscope fitted with a HCX PL APO 100 × 1.46 NA oil objective and coupled to an Andor-DU8285 VP-4094 camera. Penetration depth was 90 nm and ×2 zoom was used to obtain an optical magnification of the samples. Equal experimental conditions for acquisition were used for the different coverslips. Brightness and Contrast adjustments were equal for all images. Processing of images was performed with Image J 1.51n (Wayne Rasband national Institutes of Health, USA. http://imagej.nih.gov/ij; Java 1.8.0_66; 64 bit).

**Electron microscopy.** Adherent HEK293 cells stably transfected with control shRNA, shRab27a, or shnMNase2 were washed in PBS and fixed for 1 h in 2.5% glutaraldehyde in 0.1 M phosphate buffer at room temperature. Cells were pelleted in 15 ml Falcon tubes. Pellets were washed with phosphate buffer and incubated with 1% $OsO_4$ for 90 min at 4 °C. Samples were then dehydrated, embedded in Spurr, and sectioned on a Leica ultramicrotome (Leica Microsystems). Ultrathin sections (50–70 nm) were stained with 2% uranyl acetate for 10 min and with a lead-staining solution for 5 min and observed under a JEOL JEM-1010 transmission electron microscope fitted with a Gatan Orius SC1000 (model 832) digital camera.

For immunogold staining, cells were fixed in 4% PFA 0.5% glutaraldehyde in PBS at 4 °C. Samples were dehydrated, embedded, and sectioned. Grids were rehydrated in TBS and treated with 0.01 M glycine to deactivate reactive aldehydes. Samples were blocked with goat blocking solution (Aurion) for 1 h and incubated with primary antibodies; anti-Tfam (1:50) (Cell Signalling) and anti-DNA (1:15) (Promega) overnight in blocking solution. Grids were washed in blocking solution and incubated with secondary antibodies conjugated to 12 nm colloidal gold (1:75 for TFAM and 1:25 for DNA, Jackson ImmunoResearch) for 2 h. Samples were washed in blocking solution and distilled water. Grids were air dried for 2 h, stained with lead citrate and uranyl acetated and carbon coated for viewing under transmission electron microscope.

**Extracellular flux analysis.** Oxygen consumption rate was measured in a XF-96 Extracellular Flux Analyzer (Seahorse Bioscience) in equal numbers of shControl, shRab27a and shnSMase2 HEK293 cells incubated with XF medium. Three measurements were obtained under basal conditions and upon addition of oligomycin (1 mM), fluoro-carbonyl cyanide phenylhydrazone (FCCP; 1.5 mM), and rotenone (100 nM) + antimycin A (1 mM).

**Quantitative PCR and assessment of mitochondrial DNA by PCR.** Total RNA was isolated using the RNeasy kit (Qiagen). Genomic DNA was removed using the specific column from the kit. Reverse transcription was performed using the High Capacity cDNA Reverse Transcriptase kit (Applied Biosystems). Quantitative real-time PCR was performed in a 7900 HT Fast Real-Time PCR system (Life technologies) using SYBR green qPCR Master Mix (Promega). Data were normalized to the expression of beta-actin, beta-2-microglobulin, and/or ywhaz and analyzed with Biogazelle QBasePlus (Biogazelle). Primer sequences are specified in Supplementary Table 1.

For the analysis of mitochondrial DNA contained in exosomes, DNA from T lymphoblast-derived exosomes was isolated with the Cell Lysis Solution and Protein Precipitation Solution from Qiagen. After DNA isolation, PCR was performed, primer sequences are specified in Supplementary Table 2.

**Statistical analysis.** All values were expressed as the mean ± S.E.M. or mean with individual values included for data distribution. For in vitro experiments, statistical differences were evaluated with Student's $t$-test and between group differences were evaluated with the Mann–Whitney $U$ test for unpaired data (GraphPad Prism). Differences were considered significant when $*p < 0.05$, $**p < 0.01$, $***p < 0.001$.

**Data availability.** The authors declare that the data supporting the findings of this study are available within the article and its Supplementary Information files, or are available in a persistent repository or upon reasonable requests to the authors.

Deep-sequencing analysis data supporting the findings of this study are deposited in GEO repository with the accession number: GSE110165

DNAseq data are deposited in BioSample repository: SRP148571

Raw data from RNAseq and DNAseq analysis are under a common umbrella project: Bioproject PRJNA472779:

PRJNA432987: Mouse RNASeq data in GEO

PRJNA472354: Human DNASeq data in SRA

The mass spectrometry proteomics data are deposited to the ProteomeXchange Consortium via the PRIDE partner repository with the data set identifier PXD008843.

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

## Acknowledgements

We thank Dr. S. Bartlett for assistance with English editing and Dr A. García-Sastre for providing reagents. This study was supported by grants SAF2017/82886-R from the Spanish Ministry of Economy and Competitiveness, CAM S2017/BMD-3671 from the Comunidad de Madrid, CIBER Cardiovascular (Fondo de Investigación Sanitaria del Instituto de Salud Carlos III and co-funding by Fondo Europeo de Desarrollo Regional FEDER), ERC-2011-AdG 294340-GENTRIS and COST-Action BM1202 to F.S.-M.; grant SAF2015-65633-R from the Spanish Ministry of Economy and Competitiveness to J.A.E. M.M. is supported by MS14/00219 from Instituto de Salud Carlos III. Centro Nacional de

Investigaciones Cardiovasculares (CNIC) is supported by the Spanish Ministry of Economy and Competitiveness (MINECO) and the Pro-CNIC Foundation, and is a Severo Ochoa Center of Excellence (MINECO award SEV-2015-0505).

## Author contributions

D.T., F.B., C.V.-B., I.F.-D., A.L.-P., R.A.-P., N.B.M.-C., A.L.J., S.I., I.J.-C. performed experimental work. G.G.-A. and S.I. provided materials. F.B., D.T., J.G., M.V.-M., J.A.E., M.M. and F.S.-M. designed research. F.B., D.T., M.V.-M. and F.S.-M. analyzed data and wrote the manuscript.

## Additional information

**Competing interests:** The authors declare no competing interests.

