## [Peer Review File · Nature Communications]

Reviewers' comments:

Reviewer #1 (Immunological synapse, TCR signaling)(Remarks to the Author):

This is an interesting paper in which the authors demonstrate the presence of oxidised mitochondrial DNA on the surface of exosomes from primary T cells and Jurkat T cell line. The mtDNA can be transferred in an antigen specific manner and activates DNA sensing pathways in the target. A complication is that DNA appears to exist both on the external surface of the exosomes and within the lumen (cytoplasm). I feel that the location of the active component for the adaptive-innate sensing pathway needs to be better established and the intracellular localisation analysis required better quantification and appears it will not hold up under closer scrutiny.

1. Extracellular vesicle fractions from cell supernatants are likely to be heterogeneous mixtures. Thus its not possible to exclude the presence of one vesicle population by demonstrating the presence of molecules from another vesicle population. Thus the statement "The lack of cytochrome c and the presence of the canonical exosomal proteins CD81 and TSG101 in the purified 100 000 ×g fraction excluded the presence of apoptotic bodies in the purified EV population and suggested an endosomal origin (Figure 1d)." Also, while the presence of cytochrome C in apoptotic bodies sounds reasonable, I found no evidence for its detection in apoptotic bodies from thymocytes and other sources.
2. The statement that the exosome associate DNA was slightly reduced by DNase does not fit with the data, which shows what I would call a substantial reduction and residual DNA is degraded after lysis with Triton X-100. I think this is weak evidence that any DNA is inside the exosomes. Additional controls would be needed to make this convincing.
3. The co-localization analysis in 293 cells is not convincing and appears to be highly biased. It looks like an objective analysis with score almost all the marker as poorly co-localised and any co-localization might be accounted for by chance. The results in Figure 3 appear consistent with the endosomal and mitochondrial markers being largely distinct, as expected. The figure is a mix of data from 293 cells are various Jurkat sublines. The data that normal membrane turnover is necessary to maintain mitochondrial health is perhaps not surprising.
4. In figures 4, 5 and 6, can the authors eliminate the activity of the exosomes in transfer of mtDNA, signaling and anti-viral effects by treating them with DNase? This would help establish which compartment (vesicle surface or cytoplasmic contents) is most active in this setting. How do the authors think the exosomal DNA gets access to these cytoplasmic sensors?

Reviewer #2 (Exosome, inflammation)(Remarks to the Author):

In this study, Baixauli et al. focused on the intercellular communication between T cells and DCs via exosomes. T cells secrete exosomes containing mtDNA, which can be transferred into DCs especially through cognate interaction. The secreted exosomes affect the expression profile of IFN-stimulated genes in DCs via the STING/IRF3 pathway, which primes antiviral response in DCs. While the basic ideas presented here are certainly of interest, there are major concerns about how strongly each result would support what the authors are discussing in the paper.

The list of concerns:

1. Permeabilization procedure can change the size and characteristics of cells. Therefore, it is usually not acceptable to compare MFI between cells with or without permeabilization. It is also impossible to use the same negative control among cells with or without permeabilization. This seems to be applicable not only for cells but also for exosomes. Unless there is a strong reason that this is not true for the evaluation of exosomes, authors should not use the result shown in Fig. 2e.

2. The authors described in the text “not detected with the control isotype Ab” (page 6, line 12). However, this message seems to be not acceptable, as the expression level is compared between with anti-8-OHdG antibody and with isotype control in the experiment (Fig. 2f). Formally, the authors should perform the same experiment using parent cells and prove that oxidized DNA is preferentially packaged in exosomes, if they want to touch on “escape valve” in discussion (page 13).
3. Given the diameter of TfamGFP, most green structures look like whole mitochondria or very large mitochondrial parts in Fig. 3a. In addition, no such image is shown in Fig. 3b, in which mitochondrial components are surrounded by Rab7-positive endosomal structure. Moreover, authors should perform statistical analysis about the overlap of fluorescence in addition to representative graphs in Fig. 3a and 3f.
4. Authors should explain in detail how mitochondria changed the morphology by exosome inhibitors (Fig. 3e), ideally with statistical analysis. It is not clear to me if authors actually examined the functional change of mitochondria, although they described, “we found that inhibition of exosome biogenesis and secretion impairs mitochondrial structure and function” (page 13). Supportive data should be presented.
5. Authors concluded that macroautophagy is not involved in the secretion of exosomes including mitochondrial components (Fig. 3g and 3h). Although there is no change of the bands of ATAD3 and TFAM, it appears that CD63 or CD81 may be altered after treatment with Baflomycin or siLC3, suggesting changes in exosomal characteristics. They should at least try to respond to this concern, with reasonable argument.
6. In Fig. 4b, the bands of C57C57 in DCs do not emerge, but even disappear after addition of exosomes. This is strikingly inconsistent with the manuscript and the whole story. In addition, it is not clear to me why there is a band of C57N2B in the lane of CD4+ T cells.
7. Given the inconsistency of the results between Fig. 4b and 4e, it appears that the transfer of mtDNA may be mediated by exosomes. Authors should perform the same experiment as Fig. 4e with inhibitors of exosomes, to determine their role in mtDNA transfer.
8. The difference of results in Fig. 5d and 5f, the transcriptomic change by exosomes seems to be mediated by IRF3 but not by STING. Authors should discuss this, and delete the related sentences from the manuscript such as “exosomes did not upregulate the expression of ISGs in STING-deficient DCs” in page 9.
9. If there is no statistical significance in Supplementary Fig. 3c, the authors should not state, “DNase treatment of exosomes reduced the ability of exosomes to trigger these changes” (page 9). Even if this is true, the authors cannot conclude that mtDNA is the main player because exosomes include cellular DNA as well. Why can the authors conclude based on the shown data that “mtDNA can act transcellularly by its horizontal transfer”? (page 9).
10. In Fig. 6b, is there any significant change in gene expression between CD63GFP high and low cells? If exosomes are critically involved in the transcriptomic change in DCs, there should be clear difference here in Fig. 6b.
11. It is unclear why the authors can conclude that T cell exosomes are delivered through the IS. There remains a sufficient possibility that cognate interactions between T cells and DCs stimulate some pathways outside IS and promote secretion of exosomes. Without good reason, the authors should not say too much on the role of IS throughout this paper.

Reviewer #3 (Adaptive-innate crosstalk, T/DC function)(Remarks to the Author):

Baixauli et al examine T cell-derived exosomes, confirming that they contain mitochondrial DNA (mtDNA) as well as other components. The authors then show that in co-cultures of T cells and APCs, mtDNA can be acquired by the APC. By purifying exosomes and adding these to cultures of DC, they demonstrate substantial activation of DC and induction of interferon pathway genes. Finally, the Authors show that T cell-derived exosomes or co-culture with CD4 T cells reduces the infectivity of DCs by Vaccinia virus.

The experiments are well executed, the figures high quality and the manuscript generally well written. A major concern lies with the conclusion made in the title, abstract and in the paper that exosome transfer occurs across the immune synapse, thereby triggering DC activation. Yet, the experiments in the manuscript do not examine synaptic contacts, only co-cultures which do not distinguish when or how the T cell EVs are released or when they reach APCs. In the absence of direct examination of immune synapses, these conclusions are misleading.

Another major concern is that the experiments involve very large amounts of purified exosomes introduced into the co-cultures. It is not clear how the APC activation by large amounts (micrograms) of material compares to natural release of exosomes. It is not surprising that a large number of exosomes can activate DC. Prior studies have shown that exosomes, mitochondrial DNA, TFAM etc can activate DC (doi: 10.1038/nri.2017.21; 10.4049/jimmunol.1101375 etc). Hence these results are not novel.

Figure 6c shows that T-DC cultures reduced DC infection by vaccinia virus. Yet, these experiments do not show whether this effect was mediated by exosomes, or by other signals provided by the interactions with CD4 T cells and the antigen expressing DC (ie CD40L, cytokines).

For the experiments shown in Figure 5, what T cells and DC were used for the RNAseq analysis?

Immune contacts with T lymphocytes prime dendritic cells against pathogen infection through horizontal DNA transfer

NCOMMS-17-15516-T

We very much appreciate the constructive criticisms of the Reviewers and the opportunity to submit a revised version of our manuscript NCOMMS-17-15516-T, entitled “**Cognate immune contacts with T lymphocytes prime dendritic cells against pathogen infection through horizontal DNA transfer**” by Baixeli et al. We have taken into account all the concerns raised by the reviewers, which mainly focused on better defining the activity of exosomal DNA in triggering antiviral responses in DCs (Reviewers 1 and 2) and providing additional controls and quantifications to better support the conclusions raised on the manuscript (Reviewers 1 and 2). Following Reviewers’ suggestions, we have also clarified and discussed some of the data presented on the manuscript, and we have modified the text about the immune synapse as the subcellular location where the transfer of exosomes takes place during pathogen priming of DCs (Reviewers 2 and 3).

Please, find below a detailed point-by point reply to the Reviewers.

Reviewer #1 (Immunological synapse, TCR signaling)(Remarks to the Author):

This is an interesting paper in which the authors demonstrate the presence of oxidised mitochondrial DNA on the surface of exosomes from primary T cells and Jurkat T cell line. The mtDNA can be transferred in an antigen specific manner and activates DNA sensing pathways in the target. A complication is that DNA appears to exist both on the external surface of the exosomes and within the lumen (cytoplasm). I feel that the location of the active component for the adaptive-innate sensing pathway needs to be better established and the intracellular localisation analysis required better quantification and appears it will not hold up under closer scrutiny.

1. Extracellular vesicle fractions from cell supernatants are likely to be heterogeneous mixtures. Thus its not possible to exclude the presence of one vesicle population by demonstrating the presence of molecules from another vesicle population. Thus the statement “The lack of cytochrome c and the presence of the canonical exosomal proteins CD81 and TSG101 in the purified 100 000 xg fraction excluded the presence of apoptotic bodies in the purified EV population and suggested an endosomal origin (Figure 1d).” Also, while the presence of cytochrome C in apoptotic bodies sounds reasonable, I found no evidence for its detection in apoptotic bodies from thymocytes and other sources.

We agree with this reviewer in that the presence of exosomal proteins and the absence of cytochrome c does not exclude the possibility that other extracellular vesicles may be present in the 100 000 xg fraction. To avoid misleading the reader, the text has been modified to reflect this (page 5 in the revised manuscript). Nevertheless, the presence of mitochondrial proteins in endosomal-derived exosomes is later demonstrated by the specific silencing of the neutral sphingomyelinase 2, which inhibits exosome secretion concomitantly with a reduction in the presence of mitochondrial proteins in the 100 000 x g fraction (**Figure 3c**), and by the detection of DNA and TFAM by immunogold staining in intraluminal vesicles from canonical multivesicular bodies in the electron microscopy images (**new supplementary Figure 3b**).

2. The statement that the exosome associate DNA was slightly reduced by DNase does not fit with the data, which shows what I would call a substantial reduction and residual DNA is degraded after lysis with Triton X-100. I think this is weak evidence that any DNA is inside the exosomes. Additional controls would be needed to make this convincing.

We agree that the experimental data in Figure 2c shows a substantial reduction in DNA content upon DNase treatment. Thus, we have modified the text to represent the data more accurately (**page 6 of the revised manuscript**). Furthermore, following the Reviewer’s suggestion, we present additional evidence supporting the presence of DNA not only on the surface of exosomes but also inside them. **New Figure 2e** depicts flow cytometry analysis of permeabilized and non-permeabilized exosomes. These experiments

reveal an increase in DNA staining in exosomes upon their permeabilization (**new Figure 2e**), indicating that DNA also resides inside exosomes. Interestingly, permeabilization increases the staining of oxidized-DNA with anti-8-dGH significantly more than it increases detection of total DNA, suggesting that oxidized DNA is mostly present inside exosomes. In addition, we also provide electron microscopy images showing both DNA and TFAM by immunogold staining in intraluminal vesicles of canonical multivesicular bodies (**new Supplementary Figure 3b**), supporting our findings regarding the presence of mitochondrial DNA-associated proteins and DNA in vesicles of endosomal origin. Finally, to further demonstrate the location of DNA in exosomes, we have genetically engineered a plasmid expressing DNase II fused to the intraluminal domain of the exosome-enriched protein CD63. qPCR analysis showed that exosomes from cells expressing the CD63-DNase II chimera contain less mitochondrial DNA than exosomes from control cells (**new Supplementary Figure 4c**), indicating that mitochondrial DNA is present both inside and outside exosomes.

3. The co-localization analysis in 293 cells is not convincing and appears to be highly biased. It looks like an objective analysis with score almost all the marker as poorly co-localised and any co-localization might be accounted for by chance. The results in Figure 3 appear consistent with the endosomal and mitochondrial markers being largely distinct, as expected. The figure is a mix of data from 293 cells are various Jurkat sublines. The data that normal membrane turnover is necessary to maintain mitochondrial health is perhaps not surprising.

As the Reviewer pointed out, the overall distribution of endosomal and mitochondrial markers in cells is quite different from what we expected. Sparse co-localization spots likely represent rare events corresponding to a small fraction of the total protein, which may be important nonetheless. There are many examples of infrequent events in cells such as interorganelle contacts, like mitochondrial contacts with the endoplasmic reticulum, peroxisomes, autophagosomes or the formation of mitochondrial-derived-vesicles, which may be regulated by different stimuli and stress situations (Hailey et al., 2010; Rowland and Voeltz, 2012; Sugiura et al., 2014). The usual co-localization scores are not suitable for quantifying these types of infrequent events, which is why we used intensity histograms in selected regions of interest to demonstrate co-localization. Nevertheless, to provide an unbiased quantification of mitochondrial DNA-binding protein co-localization with early and late endosomal markers, we have increased the frequency of these events by treating the cells with low doses of FCCP (Hammerling et al., 2017). At a low dose, FCCP dissipates mitochondrial membrane potential, inducing low mitochondrial damage that favors these interactions. To compare the co-localization of mitochondrial DNA binding-proteins with early and late endosomal markers, we used Manders coefficient of the mitochondrial channel over the endosome channels (**new Figure 3a**). Using this strategy, the results show higher colocalization of the mitochondrial DNA-binding protein with late endosomal marker CD63 compared to the early endosomal marker EEA1 (also show in **new Figure 3a**). This new evidence is in agreement with our previous results showing that mitochondrial components are actively sorted into late multivesicular structures and exosomes (**Figure 3**). In order to accommodate the new functionally relevant data requested by the reviewer, Figure 3 has been reorganized and former Figures 3a and 3f have been moved to Supplementary figures 3a and 3d.

4. In figures 4, 5 and 6, can the authors eliminate the activity of the exosomes in transfer of mtDNA, signaling and anti-viral effects by treating them with DNase? This would help establish which compartment (vesicle surface or cytoplasmic contents) is most active in this setting. How do the authors think the exosomal DNA gets access to these cytoplasmic sensors?

The induction of interferon response genes is significantly reduced in recipient cells upon exosome treatment with DNase II, indicating that external DNA in exosomes is important for full induction of the anti-viral response in recipient cells (**new panel g in Figure 5**). Moreover, exosomes from cells expressing CD63-DNase II, which contain less amount of mtDNA compared with exosomes obtained from CD63-expressing control cells, also triggered a modest antiviral response in recipient cells (**new Supplementary Figure 4c and d**). All these new data support that external as well as internal exosomal DNA actively contributes to the induction of antiviral responses in recipient cells.

Regarding the mechanism through which exosomal DNA gets access to cytoplasmic sensors, we propose that the back-fusion of exosomes with endosomal membranes may release their content into the cytoplasm and activate cytoplasmic DNA innate immune sensors. This mechanism has been proposed to explain how exosomes may release their content into recipient cells (Mulcahy et al., 2014). However, our results show that the contribution of cytoplasmic sensors, e.g. STING, is only partial (**Figure 5d**). This means that other DNA receptors localized at the endosome or the plasma membrane could also sense the DNA present on the surface of exosomes and activate interferon-response factors, thus contributing to the final induction of a potent anti-viral response. Discussion on this topic has now been added to the text (**page 13 of the revised manuscript**).

Reviewer #2 (Exosome, inflammation)(Remarks to the Author):

In this study, Baixauli et al. focused on the intercellular communication between T cells and DCs via exosomes. T cells secrete exosomes containing mtDNA, which can be transferred into DCs especially through cognate interaction. The secreted exosomes affect the expression profile of IFN-stimulated genes in DCs via the STING/IRF3 pathway, which primes antiviral response in DCs. While the basic ideas presented here are certainly of interest, there are major concerns about how strongly each result would support what the authors are discussing in the paper.

The list of concerns:

1. Permeabilization procedure can change the size and characteristics of cells. Therefore, it is usually not acceptable to compare MFI between cells with or without permeabilization. It is also impossible to use the same negative control among cells with or without permeabilization. This seems to be applicable not only for cells but also for exosomes. Unless there is a strong reason that this is not true for the evaluation of exosomes, authors should not use the result shown in Fig. 2e.

We agree with the Reviewer; permeabilization may change some exosomal features, possibly affecting MFI. To redress this, we have included negative controls with permeabilized and non-permeabilized exosomes in **new Figure 2e**. As shown there, flow cytometry analysis of permeabilized and non-permeabilized exosomes reveals an increase in total DNA staining in exosomes upon their permeabilization (**new Figure 2e**), indicating that DNA is present both outside and inside exosomes. Interestingly, although the staining of oxidized-DNA with anti-8-dGH antibody increases in both (permeabilized and non-permeabilized exosomes) with respect to negative controls, it increases significantly more than total DNA detection upon permeabilization, suggesting that oxidized DNA is mostly present inside exosomes.

2. The authors described in the text “not detected with the control isotype Ab” (page 6, line 12). However, this message seems to be not acceptable, as the expression level is compared between with anti-8-OHdG antibody and with isotype control in the experiment (Fig. 2f). Formally, the authors should perform the same experiment using parent cells and prove that oxidized DNA is preferentially packaged in exosomes, if they want to touch on “escape valve” in discussion (page 13).

Following the Reviewer’s suggestion, we have performed immunoprecipitation experiments with anti-8-OHdG and anti-total-DNA in exosomes and in mitochondria-enriched fractions of parental cells. qPCR analysis of all mitochondrial genes analyzed and quantification of the ratio between oxidized DNA/total DNA in parental cells and exosomes reveals an enrichment of oxidized mitochondrial DNA in the exosome fraction compared to parental cells (**new Figure 2f**), indicating that oxidized mitochondrial DNA is preferentially packaged into exosomes. We appreciate the reviewer for proposing this experiment, which reveals in a clearer manner the secretion of oxidized mitochondrial DNA in exosomes and supports the conclusion that oxidized mitochondrial DNA is preferentially packaged into exosomes.

3. Given the diameter of TfamGFP, most green structures look like whole mitochondria or very large mitochondrial parts in Fig. 3a. In addition, no such image is shown in Fig. 3b, in which mitochondrial components are surrounded by Rab7-positive endosomal structure. Moreover, authors should perform statistical analysis about the overlap of fluorescence in addition to representative graphs in Fig. 3a and 3f.

We agree in that overexpression of Tfam-GFP seems to decorate very large mitochondrial parts, an effect that could be due to excessive expression of exogenous Tfam-GFP. To avoid misinterpretation of these images due to exogenous Tfam protein overexpression, we have repeated the experiments labeling an endogenous mitochondrial DNA binding protein, single-stranded DNA-binding protein 1 (SSBP1) together with early (EEA1) and late endosomal markers (CD63). As presented in **new Figure 3a**, immunofluorescence images and the accompanying quantification indicate a higher co-localization of mitochondrial proteins with late endosomal markers than with early endosomal markers, supporting our conclusion that mitochondrial proteins segregate into late endosomal compartments. The new Figure 3 has been reorganized to accommodate new functionally relevant data as requested by the reviewer, thus former Figures 3a and 3f have been moved to Supplementary figures 3a and 3d.

4. Authors should explain in detail how mitochondria changed the morphology by exosome inhibitors (Fig. 3e), ideally with statistical analysis. It is not clear to me if authors actually examined the functional change of mitochondria, although they described, “we found that inhibition of exosome biogenesis and secretion impairs mitochondrial structure and function” (page 13). Supportive data should be presented.

Inhibition of exosome biogenesis and secretion by the specific silencing of Rab27a or nSMase2 changed the morphology of mitochondria by increasing significantly mitochondrial cristae width, as quantified in the electron microscopy images (**new Figure 3e**). The increase in cristae width has been previously related to changes in mitochondrial electron transport chain complex organization that in turn may result in functional changes in cellular respiration ability (Cogliati et al., 2013). In accordance, inhibition of exosome secretion by Rab27a or nSMase2 silencing significantly decreased mitochondrial respiration in these cells, measured by Seahorse (**new Figure 3f**). These data support a role for exosomes in sustaining mitochondrial function by secretion of mitochondrial components.

5. Authors concluded that macroautophagy is not involved in the secretion of exosomes including mitochondrial components (Fig. 3g and 3h). Although there is no change of the bands of ATAD3 and TFAM, it appears that CD63 or CD81 may be altered after treatment with Bafilomycin or siLC3, suggesting changes in exosomal characteristics. They should at least try to respond to this concern, with reasonable argument.

The decrease in CD63 secretion upon bafilomycin treatment was not representative of other experiments and might have been due to technical problems in Western blotting and not to a real decrease in protein levels. Therefore, this Western blot has been replaced with a more representative replicate (**new Figure 3g**). Moreover, to rule out the effect of these treatments on exosomal features, we also include Nanoparticle tracking analyses of size and concentration of extracellular vesicles in these conditions (**please see new Figure 3g and 3h and Supplementary Figure 3e,f**). These new results rule out any effect on exosome size, indicating a possible defect in exosome secretion upon starvation conditions. In any case, a slight increase in the secretion of exosome markers can be sometimes observed upon bafilomycin treatment, which may reflect a mechanism to compensate the lack of degradation of certain endosomal components in lysosomes by increasing their release through extracellular vesicles. In addition, it should be taken into account that CD63 and CD81 can be present in other extracellular vesicles apart from exosomes, such as membrane-derived vesicles; therefore, not all components present in the 100 000 x g pellet always change in the same extent upon different stimuli.

6. In Fig. 4b, the bands of C57C57 in DCs do not emerge, but even disappear after addition of exosomes. This is strikingly inconsistent with the manuscript and the whole story. In addition, it is not clear to me why there is a band of C57N2B in the lane of CD4+ T cells.

We apologize for the lack of clarity regarding the experiment presented in Figure 4b. Although the intensity of the bands is low, the bands corresponding to C57C57 mitochondrial DNA appear in C57N2B DCs upon addition of exosomes (lanes 1/3, 2/3 and 3/3 with respect to lane 0, indicated by labelled arrows). The intensity of the fragmented PCR bands after digestion is represented on the histograms beside and the arrows and they indicate the slopes measured in the conditions in which exosomes are present. We hope the changes done to the figure improve its clarity.

Regarding the presence of a band at the same height than C57N2B in the lane of CD4 T cells, this is due to uncompleted digestion of PCR DNA in the RFLP analysis; the efficiency of the enzymatic digestion is not always 100%, especially if the starting material is very abundant.

7. Given the inconsistency of the results between Fig. 4b and 4e, it appears that the transfer of mtDNA may be mediated by exosomes. Authors should perform the same experiment as Fig. 4e with inhibitors of exosomes, to determine their role in mtDNA transfer.

Our experiments show the transfer of mtDNA from C57C57 T cells to NZB DCs upon addition of T cell exosomes to DCs (Fig. 4b) or in T cell-DC co-cultures upon antigen recognition (Fig. 4e). In addition, our infection experiments with vaccinia in the context of antigen-dependent contacts with T cells pre-treated with manumycin (an inhibitor of exosome secretion) provide a functional demonstration of the role of exosomes in priming antiviral responses in recipient DCs (Fig. 6e).

8. The difference of results in Fig. 5d and 5f, the transcriptomic change by exosomes seems to be mediated by IRF3 but not by STING. Authors should discuss this, and delete the related sentences from the manuscript such as “exosomes did not upregulate the expression of ISGs in STING-deficient DCs” in page 9.

We agree with the Reviewer that data presented in Figure 5d do not allow to unequivocally demonstrate that exosomes do not upregulate the expression of interferon stimulated genes (ISGs) in STING-deficient cells. Therefore, the text has been modified to better describe the data (**please see page 10 of the revised manuscript**). However, statistical analysis reveals that the upregulation of ISGs upon exosome treatment is significantly reduced in STING-deficient recipient cells with respect to wild-type cells, supporting that, although other DNA sensors might be involved, this response is partially mediated through STING. Discussion on this part has been added on **page 13** of the revised manuscript.

9. If there is no statistical significance in Supplementary Fig. 3c, the authors should not state, “DNase treatment of exosomes reduced the ability of exosomes to trigger these changes” (page 9). Even if this is true, the authors cannot conclude that mtDNA is the main player because exosomes include cellular DNA as well. Why can the authors conclude based on the shown data that “mtDNA can act transcellularly by its horizontal transfer”? (page 9).

We have repeated these experiments with a different DNase treatment protocol that enables a less aggressive DNase inactivation in the samples, allowing less interference on the response measured in recipient DCs. To better compare both situations, we have used the same protocol in control exosomes with the only difference being the presence of the DNase. Using these conditions, the induction of antiviral genes is clearly and significantly reduced upon DNase treatment, indicating that antiviral priming is mediated by exosomal DNA (**new Figure 5g**). As pointed out by the reviewer, this experiment does not exclude the possibility that other DNA of non-mitochondrial origin may be contained in exosomes and contribute to this response. This has been clearly stated in the text (please see **pages 10 and 13** of the revised manuscript).

10. In Fig. 6b, is there any significant change in gene expression between CD63GFP high and low cells? If exosomes are critically involved in the transcriptomic change in DCs, there should be clear difference here in Fig. 6b.

Under a very stringent sorting/separation conditions, we were able to see a clear upregulation of CXCL10 in CD63GFP HIGH with respect to CD63GFP LOW recipient cells (**Figure 6b**). Although a tendency is observed with other genes, the changes are not significant and we have omitted the other genes from the analysis in this panel. The differences obtained using this model may rely on the different cell types used. These experiments were performed with a human model based on Jurkat J77 (T) and Raji (B) cell lines. The reason for using cells lines is that overexpression of CD63GFP in primary T cells is technically very difficult. Moreover, it is important to consider that, in addition to the use of cell lines and the different origin (human), in this model of cognate immune interactions, the recipient cell is a B cell line and not a primary DC, which could partially explain the lower differences found in antiviral gene upregulation in these cells. Nevertheless, the importance of exosomes in the generation of an antiviral response in antigen presenting

cells upon antigen cognate contacts with T cells is later demonstrated with exosome inhibitors in a mouse model with primary cells (**Figure 6e**).

11. It is unclear why the authors can conclude that T cell exosomes are delivered through the IS. There remains a sufficient possibility that cognate interactions between T cells and DCs stimulate some pathways outside IS and promote secretion of exosomes. Without good reason, the authors should not say too much on the role of IS throughout this paper.

Our group already reported the specific transfer of T cell exosomes through the immune synapse (IS) (Mittelbrunn et al., 2011). Specifically, we demonstrated the polarization of MVBs towards the IS in T cells and the confined and unidirectional transfer of exosomes from T cells to antigen-presenting cells, which is dependent on antigen recognition (Mittelbrunn et al., 2011). Michael Dustin's group and collaborators have also shown the antigen-dependent secretion of extracellular vesicles at the center of the IS (Choudhuri et al., 2014), supporting the notion that the IS acts as a really efficient device for the transfer of molecules and vesicles between these two cells. In this work, our data show that antigen recognition is required for the transfer of mitochondrial components from T cells to DCs (**Figure 4**), although it is true that the experiments presented here do not specifically demonstrate that the transfer of mitochondrial DNA-bearing exosomes takes place within and across the IS. **This has been clearly stated now throughout the text and the title and abstract modified in accordance.**

Reviewer #3 (Adaptive-innate crosstalk, T/DC function)(Remarks to the Author):

Baixauli et al examine T cell-derived exosomes, confirming that they contain mitochondrial DNA (mtDNA) as well as other components. The authors then show that in co-cultures of T cells and APCs, mtDNA can be acquired by the APC. By purifying exosomes and adding these to cultures of DC, they demonstrate substantial activation of DC and induction of interferon pathway genes. Finally, the Authors show that T cell-derived exosomes or co-culture with CD4 T cells reduces the infectivity of DCs by Vaccinia virus.

The experiments are well executed, the figures high quality and the manuscript generally well written. A major concern lies with the conclusion made in the title, abstract and in the paper that exosome transfer occurs across the immune synapse, thereby triggering DC activation. Yet, the experiments in the manuscript do not examine synaptic contacts, only co-cultures which do not distinguish when or how the T cell EVs are released or when they reach APCs. In the absence of direct examination of immune synapses, these conclusions are misleading.

Previous reports have shown the polarization of MVB towards the IS in T cells and the transfer of exosomes from T cells to antigen-presenting cells, which is dependent on antigen recognition (Mittelbrunn et al., 2011). Later studies have also shown antigen-dependent secretion of extracellular vesicles at the center of the IS (Choudhuri et al., 2014). Nevertheless, although we show that antigen recognition is required, it is true that experiments presented here do not specifically demonstrate that the transfer of mitochondrial DNA-bearing exosomes takes place across the IS. **This has been clearly stated in the text, and the title and abstract modified in accordance.**

Another major concern is that the experiments involve very large amounts of purified exosomes introduced into the co-cultures. It is not clear how the APC activation by large amounts (micrograms) of material compares to natural release of exosomes. It is not surprising that a large number of exosomes can activate DC. Prior studies have shown that exosomes, mitochondrial DNA, TFAM etc can activate DC (doi: 10.1038/nri.2017.21; 10.4049/jimmunol.1101375 etc). Hence these results are not novel. Figure 6c shows that T-DC cultures reduced DC infection by vaccinia virus. Yet, these experiments do not show whether this effect was mediated by exosomes, or by other signals provided by the interactions with CD4 T cells and the antigen expressing DC (ie CD40L, cytokines).

Our experiments show that, upon co-culture with T cells and in an antigen recognition-dependent manner, DCs have increased protection against vaccinia infection (**Figure 6d**), and this protection is lost when exosome secretion is inhibited by manumycin treatment (**Figure 6e**). Therefore, we do not only

demonstrate that purified exosomes induce an anti-viral response in DCs, but also that exosomes physiologically released in T cell-DC co-cultures upon antigen recognition prime antiviral response in DCs.

For the experiments shown in Figure 5, what T cells and DC were used for the RNAseq analysis?

For these experiments we used mouse DCs differentiated from bone marrow precursors and T cells obtained from mouse splenocytes.

References

1. Choudhuri, K., Llodra, J., Roth, E.W., Tsai, J., Gordo, S., Wucherpfennig, K.W., Kam, L.C., Stokes, D.L., and Dustin, M.L. (2014). Polarized release of T-cell-receptor-enriched microvesicles at the immunological synapse. *Nature* **507**, 118-123.
2. Cogliati, S., Frezza, C., Soriano, M.E., Varanita, T., Quintana-Cabrera, R., Corrado, M., Cipolat, S., Costa, V., Casarin, A., Gomes, L.C., *et al.* (2013). Mitochondrial cristae shape determines respiratory chain supercomplexes assembly and respiratory efficiency. *Cell* **155**, 160-171.
3. Hailey, D.W., Rambold, A.S., Satpute-Krishnan, P., Mitra, K., Sougrat, R., Kim, P.K., and Lippincott-Schwartz, J. (2010). Mitochondria supply membranes for autophagosome biogenesis during starvation. *Cell* **141**, 656-667.
4. Hammerling, B.C., Najor, R.H., Cortez, M.Q., Shires, S.E., Leon, L.J., Gonzalez, E.R., Boassa, D., Phan, S., Thor, A., Jimenez, R.E., *et al.* (2017). A Rab5 endosomal pathway mediates Parkin-dependent mitochondrial clearance. *Nat Commun* **8**, 14050.
5. Mittelbrunn, M., Gutierrez-Vazquez, C., Villarroya-Beltri, C., Gonzalez, S., Sanchez-Cabo, F., Gonzalez, M.A., Bernad, A., and Sanchez-Madrid, F. (2011). Unidirectional transfer of microRNA-loaded exosomes from T cells to antigen-presenting cells. *Nat Commun* **2**, 282.
6. Mulcahy, L.A., Pink, R.C., and Carter, D.R. (2014). Routes and mechanisms of extracellular vesicle uptake. *J Extracell Vesicles* **3**.
7. Rowland, A.A., and Voeltz, G.K. (2012). Endoplasmic reticulum-mitochondria contacts: function of the junction. *Nat Rev Mol Cell Biol* **13**, 607-625.
8. Sugiura, A., McLelland, G.L., Fon, E.A., and McBride, H.M. (2014). A new pathway for mitochondrial quality control: mitochondrial-derived vesicles. *EMBO J* **33**, 2142-2156.

Reviewers' comments:

Reviewer #1 (Remarks to the Author):

The study suggests that T cell will induce an anti-viral state in DC independent of the context of the interaction due to transfer of exosomes associated both inside and out with oxidized mitochondrial DNA that engaged multiple innate sensing mechanisms in DC. At the end of the day, they are taking a specific size range of subcellular debris, including exosomes, but probably many other small cell fragments, and exposing DC to high concentrations of these particles- leading to a response to the associated DNA. The new data emphasize these limitations and prove that at least the oxidized DNA is in a subset of particles studied. When these compartments are designed to incorporate DNase then the activity of the DNA to stimulate DC is reduced. There are still conceptual and technical issues that need to be addressed.

The term horizontal DNA or gene transfer is used in genetics to distinguish inheritance from parent to progeny (vertical) vs inheritance from one individual to another within or across species (horizontal or lateral). The term horizontal transfer has been stretched in recent years to also mean transfer of DNA between cells (particularly cancer cells) to healthy surrounding cells that leads to integration or inheritance as an episome capable of supporting gene expression in progeny of the healthy cell- potentially transforming it. In this manuscript the same exact phrase is used to mean something entirely different, which is just the transfer of fragments of DNA is extended to innate sensing of DNA. So this completely removes the term from the context of inheritance and gene expression into another functional setting- triggering innate sensing as a sort of damage associated molecules pattern. I feel that it's quite confusing and the authors should just refer to it as DNA transfer as the distinction between horizontal and vertical seems not to apply here as there is no evidence of any inheritance process.

The authors have not taken on board the criticism that they have a mixed population of vesicles that may contain exosomes and many other particles that will co-fractionate. They would need to do some single particle analysis showing that the exosome and mitochondrial markers are in the same vesicles to really say what particles contain or associate with DNA. So they can't say that the relevant particles are exosomes.

The flow cytometry in Fig 2e for oxidized DNA in permeabilized vesicles suggests that this signal is coming from a small subset of vesicles where each vesicle has a large signal. Typically these capture methods collect many exosomes per bead and the signals represent the average of many vesicles needed to get a signal and just provide a population average that reveals no single particle behavior or population heterogeneity. In this instance, there is strong sub-population of around 1/3rd containing the oxidized DNA. That would suggest that a subset of the aldehyde beads have a single vesicle coupled to them containing a large amount of oxidized DNA. This is very strong support for a rare EV subset containing the material of greatest interest to the study and does not support the notion that the relevant material is in exosomes.

The studies in the 293 cells looking at subcellular localization are not convincing and it's not clear that they are that relevant in the end as the particles studied in the context of DC are from T cells. If this is a conserved mechanism in all cells then the lymph node environment will be laced with these particles and the DC will need to be desensitized to these in the steady state to be able to mediate tolerance induction or induce an immune response in the presence of a pathogen.

Reviewer #2 (Remarks to the Author):

The authors have mostly responded to the concerns raised in my previous comments. I just have one concern that needs to be addressed.

Regarding the concern no. 6 in my previous comment:

May I understand that the band at the same height as NZB in the lane of CD4+ T cells (Fig. 4b, Lower) is not NZB, but is an artifact due to uncompleted digestion of DNA? If this is the case, this figure misleads the readers, and authors should ideally replace this figure, or at least address the phenomenon clearly both in the main text and the figure legend. In addition, the figure of band intensity (lower, right) is still a bit problematic. I think the authors want the readers to find small peaks pointed by arrows in the samples in the presence of exosomes. But the intensity slope in the sample in the absence of exosomes is higher than the samples in the presence of exosomes. The authors should discuss on this point.

Reviewer #3 (Remarks to the Author):

The Authors have mostly answered my questions, although in regards to Figure 6E, it needs to be demonstrated that pretreatment of the OT-II T cells does not affect their viability throughout the co-culture, due to the potentially toxic effects of this compound. It is critical for the conclusions of the study to show that this effect was mediated by exosomes, and not off target effects such as T cell death or an inability of the T cells to respond under increasing concentrations of manumycin A

Immune contacts with T lymphocytes prime dendritic cells against pathogen infection through transfer of exosomal DNA

NCOMMS-17-15516-T

We very much appreciate the remaining minor comments of the Reviewers and the opportunity to submit a revised version of our manuscript NCOMMS-17-15516-T, entitled “**Cognate immune contacts with T lymphocytes prime dendritic cells against pathogen infection through horizontal DNA transfer**” by Baixauli et al. We have taken into account all the remaining concerns raised by the reviewers.

Reviewer #1

The study suggests that T cell will induce an anti-viral state in DC independent of the context of the interaction due to transfer of exosomes associated both inside and out with oxidized mitochondrial DNA that engaged multiple innate sensing mechanisms in DC. At the end of the day, they are taking a specific size range of subcellular debris, including exosomes, but probably many other small cell fragments, and exposing DC to high concentrations of these particles- leading to a response to the associated DNA. The new data emphasize these limitations and prove that the at least the oxidized DNA is in a subset of particles studied. When these compartments are designed to incorporate DNase then the activity of the DNA to stimulate DC is reduced. There are still conceptual and technical issues that need to be addressed.

The term horizontal DNA or gene transfer is used in genetics to distinguish inheritance from parent to progeny (vertical) vs inheritance from one individuals to another within or across species (horizontal or lateral). The term horizontal transfer has been stretched in recent years to also mean transfer of DNA between cells (particularly cancer cells) to healthy surrounding cells that leads to integration or inheritance as an episome capable of supporting gene expression in progeny of the healthy cell- potentially transforming it. In this manuscript the same exact phrase is used to mean something entirely different, which is just the transfer of fragments of DNA is extended to innate sensing of DNA. So this completely removes the term from the context of inheritance and gene expression into another functional setting- triggering innate sensing as a sort of damage associated molecules pattern. I feel that its quite confusing and the authors should just refer to it as DNA transfer as the distinction between horizontal and vertical seems not to apply here as there is no evidence of any inheritance process.

Following the reviewer’s suggestion, we have omitted the word “horizontal” from “Horizontal transfer”, using instead the term “transfer” alone. This has been modified throughout the text of the revised manuscript.

The authors have not taken on board the criticism that they have a mixed population of vesicles that may contain exosomes and many other particles that will co-fractionate. They would need to do some single particle analysis showing that the exosome and mitochondrial markers are in the same vesicles to really say what particles contain or associate with DNA. So they can’t say that the relevant particles are exosomes.

Although this concern was not explicitly brought up in the previous critique of the reviewer, we agree that we cannot completely rule out that exosomal proteins and DNA are in the same vesicles. The co-fractionation data using sucrose gradient ultracentrifugation of exosomal markers with mtDNA-binding proteins (Fig 1e) and DNA (Fig. 2d), together with the inhibition of their secretion by knocking-down the nSMase 2 (Fig. 3c), support that mtDNA and mtDNA-binding proteins are harbored and loaded into exosomes. Nevertheless, we have taken this new suggestion into account, and we have performed additional control experiments, which now appear in Fig. 2e and Supplementary Fig. 2a. Specifically, we have isolated

the exosomal fraction obtained from T cells, adhered this fraction to aldehyde sulphate beads and co-stained them with anti-CD81 and anti-DNA. Flow cytometry analysis show that the majority of aldehyde sulphate beads are positive for both DNA and CD81 markers, (Fig. 2e, and Supplementary Fig. 2a), supporting the presence of DNA in exosomes.

The flow cytometry in Fig 2e for oxidized DNA in permeabilized vesicles suggests that this signal is coming from a small subset of vesicles where each vesicle has a large signal. Typically these capture methods collect many exosomes per bead and the signals represent the average of many vesicles needed to get a signal and just provide a population average that reveals no single particle behavior or population heterogeneity. In this instance, there is strong sub-population of around 1/3rd containing the oxidized DNA. That would suggest that a subset of the aldehyde beads have a single vesicle coupled to them containing a large amount of oxidized DNA. This is very strong support for a rare EV subset containing the material of greatest interest to the study and does not support the notion that the relevant material is in exosomes.

- We thank the reviewer for this suggestion. When addressing this issue by using a number of different permeabilization procedures, we have detected that different permeabilization/ fixation protocols alter the recognition of the anti-oxidized DNA antibody, giving a high variability in the results. Conversely, this was not the case under non-permeabilizing conditions. For the sake of consistency, we have omitted the data showing reactivity of this antibody with exosomes upon permeabilization. Nevertheless, this does not affect our main conclusion from this Figure since the data obtained in the chromatin immunoprecipitation experiments with the anti-oxDNA antibody support the notion that oxidized mtDNA is preferentially packaged into extracellular vesicles compared with parental cells.

The studies in the 293 cells looking at subcellular localization are not convincing and its not clear that they are that relevant in the end as the particles studied in the context of DC are from T cells. If this is a conserved mechanism in all cells then the lymph node environment will be laced with these particles and the DC will need to be desensitized to these in the steady state to be able to mediate tolerance induction or induce an immune response in the presence of a pathogen.

In this work, we present evidence of the role of T cell-APC contacts as conductors and facilitators of the transfer of T cells exosomes loaded with mtDNA and mtDNA-binding proteins to DCs. Although mitochondrial content does not only appear in exosomes from immune cells, our work reveals that T cells boost inflammatory responses in DCs in the context of immune-dependent contacts. Specifically, we demonstrate that T cells boost antiviral responses in DCs by transferring immunogenic molecules packaged in exosomes through the antigen-driven contacts formed between T cells and DCs, which supports an undescribed role for the these antigen-driven cell-to-cell contacts as devices to boost antiviral responses in DCs. As for the use of 293 cells, the reviewer is likely aware of the size difference between these cells and T cells, which facilitated the co-localization experiments. In this manner, these experiments strengthen our findings regarding the loading and secretion of mitochondrial components not restricted to immune cells, becoming a mechanism used by several cell types to secrete mitochondrial components.

Reviewer 2

The authors have mostly responded to the concerns raised in my previous comments. I just have one concern that needs to be addressed. Regarding the concern no. 6 in my previous comment: May I understand that the band at the same height as NZB in the lane of CD4+ T cells (Fig. 4b, Lower) is not NZB, but is an artifact due to uncompleted digestion of DNA? If this is the case, this figure misleads the readers, and authors should ideally replace this figure, or at least address the phenomenon clearly both in the main text and the figure legend. In addition, the figure of band intensity (lower, right) is still a bit problematic. I think the authors want the readers to find small peaks pointed by arrows in the samples in the presence of exosomes. But the intensity slope in the sample in the absence of exosomes is higher than the samples in the presence of exosomes. The authors should discuss on this point.

To avoid misleading of the readers, we have omitted panel b, lower part from the figure. We think that deleting this panel from the manuscript does not affect the main conclusion of this Figure, which demonstrates the unidirectional transfer of mtDNA between immune cells during immune cognate interactions.

Reviewer 3

The Authors have mostly answered my questions, although in regards to Figure 6E, it needs to be demonstrated that pretreatment of the OT-II T cells does not affect their viability throughout the co-culture, due to the potentially toxic effects of this compound. It is critical for the conclusions of the study to show that this effect was mediated by exosomes, and not off target effects such as T cell death or an inability of the T cells to respond under increasing concentrations of manumycin A.

As requested by Reviewer 3, additional controls of cell viability and T cell activation with different doses of manumycin are now included under Supplementary Fig. 5. These new results indicate that the doses employed of manumycin A do not affect T cell viability and activation, supporting our results showing the activation of antiviral response genes during antigen-dependent contacts is mediated through T cell exosomes.

Reviewers' comments:

Reviewer #1 (Remarks to the Author):

The authors have clarified the terminology and removed horizontal to avoid confusing. The issue of whether the mitochondrial DNA is really in exosomes or just a co-sedimenting structure that partly co-sediments with exosomes is not addressed. Addressing this issue would really require that the authors perform some single particle analysis of the particles that are isolating. Its technically possible to do this by flow cytometry with some modifications or specialised instruments, or by single particle imaging. The experiment where thousands of particles are attached to beads and then run through a flow cytometer provides no evidence that these are in the vesicles with the exosomal markers. In fact, the density gradient in Fig 2D clearly shows that the mtDNA is in different structures than the exosome markers as the peak and tail locations are different. So at best the DNA is in a subset and more likely in a different population.

I agree with reviewer 2 that there is a weakness in experiment in Fig 2 c,d and e. In the co-cultures the DC can just engulf the T cells whereas T cells are not likely to engulf DC. This could reasonably be antigen driven as antigen recognition would give the DC close access to the T cell for long periods. Its consistent with their expectations, but its a weak experiment.

The balance of evidence supports DC being activated by something from the T cells and this is consistent with innate sensing of mitochondrial DNA based on other work that mito DNA is a DAMP, and they find this is the extracellular debris, but its not clear if this is really exosomes vs some other type of subcellular/submitochondrial particles.

Reviewer #2 (Remarks to the Author):

The authors omitted the lower part of Fig. 4b in the revised manuscript. However, the experiment is a key component contributing to their conclusion of this study. The original result (Fig. 4b, lower part) described that mtDNA can be transferred via exosomes, and this is fundamental to conclude that exosomal transfer of mtDNA is critical in the T-DC interaction. The authors should respond to my concerns about this figure that I clarified in the previous comment.

Reviewer #3 (Remarks to the Author):

To ensure that manumycin A treatment of T cells was not affecting viability or activation, thereby supporting the conclusions of the study, I requested that the experiment in Figure 6e was repeated to look at OT-II cell viability and activation. In Figure 6e OT-II T cells were pre-treated with manumycin (for how long?; it is not stated) and then incubated for 4hr with OVA peptide loaded DC before infection with vaccinia-GFP and analysis. The data show a 20% reduction in infected DC, that is abrogated with 2.5uM manumycin A pre-treatment (but not with 1uM). In new data shown in Suppl Fig5, T cells (it is not stated if these are bulk T cells, CD4 T cells or OT-II T cells) were activated with anti-CD3 and anti-CD28 with concurrent manumycin A treatment. In this experiment, it appears that roughly 20% of the cells were non-viable. This could explain the reduction in protection observed in Figure 6e. Since at most 3% of the DCs in culture were infected during the assay, a loss of up to 20% of the T cells due to death or dysfunction may explain, at least in part, a reduction in ability to influence DC infection in the culture. This does not support the papers conclusions.

Replies to Reviewers Comments

Reviewer 1.

The authors have clarified the terminology and removed horizontal to avoid confusing. The issue of whether the mitochondrial DNA is really in exosomes or just a co-sedimenting structure that partly co-sediments with exosomes is not addressed. Addressing this issue would really require that the authors perform some single particle analysis of the particles that are isolating. Its technically possible to do this by flow cytometry with some modifications or specialised instruments, or by single particle imaging. The experiment where thousands of particles are attached to beads and then run through a flow cytometer provides no evidence that these are in the vesicles with the exosomal markers. In fact, the density gradient in Fig 2D clearly shows that the mtDNA is in different structures than the exosome markers as the peak and tail locations are different. So at best the DNA is in a subset and more likely in a different population. I agree with reviewer 2 that there is a weakness in experiment in Fig 2 c,d and e. In the co-cultures the DC can just engulf the T cells whereas T cells are not likely to engulf DC. This could reasonably be antigen driven as antigen recognition would give the DC close access to the T cell for long periods. Its consistent with their expectations, but its a weak experiment. The balance of evidence supports DC being activated by something from the T cells and this is consistent with innate sensing of mitochondrial DNA based on other work that mito DNA is a DAMP, and they find this is the extracellular debris, but its not clear if this is really exosomes vs some other type of subcellular/submitochondrial particles.

-- Performing these experiments would require methodologies technically unfeasible in our facilities, e.g. STORM. We have attempted to address experimentally this issue using TIRF, currently available in our Microscopy Facility. Although the resolution level of these methodologies does not render unequivocal conclusions, we have obtained new data, which we have added as new Supplementary Fig. 2b, and the corresponding text under materials and Methods (Pages 33-34) and Legend to Supp Fig. 2b. The new information points out that DNA is not present in all CD81-positive EVs, but they both co-localize in a small subset of CD81⁺-exosomes containing DNA. Nevertheless, this does not represent a significant biological difference in the main message of the manuscript, which was demonstrated by the loss of function data obtained with the DNase or CD63-DNase fusion proteins (Figures 5g and Supp 4c,d). As suggested by the reviewer and the editor, we have clarified this issue and toned down our statements throughout the text (page 6, first paragraph).

Reviewer 2

The authors omitted the lower part of Fig. 4b in the revised manuscript. However, the experiment is a key component contributing to their conclusion of this study. The original result

(Fig. 4b, lower part) described that mtDNA can be transferred via exosomes, and this is fundamental to conclude that exosomal transfer of mtDNA is critical in the T-DC interaction. The authors should respond to my concerns about this figure that I clarified in the previous comment.

--Thanks for your input. Following your request, we have returned the lower part of Fig. 4b to the manuscript. As indicated, we have discussed the caveat of the data in the Figure legend and text, that is, the partial digestion of C57 DNA observed in CD4 cells (Page 8, last paragraph and page 27 last sentence).

Reviewer 3

To ensure that manumycin A treatment of T cells was not affecting viability or activation, thereby supporting the conclusions of the study, I requested that the experiment in Figure 6e was repeated to look at OT-II cell viability and activation. In Figure 6e OT-II T cells were pre-treated with manumycin (for how long?; it is not stated) and then incubated for 4hr with OVA peptide loaded DC before infection with vaccinia-GFP and analysis. The data show a 20% reduction in infected DC, that is abrogated with 2.5uM manumycin A pre-treatment (but not with 1uM). In new data shown in Suppl Fig5, T cells (it is not stated if these are bulk T cells, CD4 T cells or OT-II T cells) were activated with anti-CD3 and anti-CD28 with concurrent manumycin A treatment. In this experiment, it appears that roughly 20% of the cells were non-viable. This could explain the reduction in protection observed in Figure 6e. Since at most 3% of the DCs in culture were infected during the assay, a loss of up to 20% of the T cells due to death or dysfunction may explain, at least in part, a reduction in ability to influence DC infection in the culture. This does not support the papers conclusions.

-- In the final version, we provide all the details in the text corresponding to Fig. 6e and Supp Fig. 5a under Materials and Methods (Page 32, under subheading "Cell viability assay") and in the Figure legends, as requested. We have included data with CD4 T lymphoblasts activated with anti-CD3/CD28, or with OVA-pulsed dendritic cells (DC) (New panel in Supplementary figure 5a). Furthermore, it is important to clarify that the calculated percentages of cell death in manumycin-treated cells stated does not accurately reflect the data. The mean of cell death in four independent experiments (CD3/CD28) and three independent experiments in T cells activated with OVA-DC is approximately 13% for maximal concentration of the drug, which is not statistically significant. In fact, taking cell death into account only reduces the ratio of T:DC from 5:1 to 4.35:1 in the assay, ensuring that functional T cells will still be present in the co-culture system. Therefore, such a modest degree of cell death would not explain the complete loss of antiviral protection by manumycin.

REVIEWERS' COMMENTS:

Reviewer #1 (Remarks to the Author):

The data in Figure S2 was exactly what I was hoping for regarding single particle analysis. The data suggests that all CD81+ structures are TSG101+ and that all DNA+ structures also have some level of CD81, but they may be somewhat anti-correlated in terms of brightness. A plot of single particle fluorescent signals for CD81 vs TSG101 and CD81 vs DNA would provide sufficient information for anyone trying to reproduce this work as a key analytical tool to determine if they have prepared the material correctly. Thus, I would only propose that the authors provide a scatter plot of >100 vesicles, which should be minimally sufficient to demonstrate the frequency of particles that are CD81^{high} DNA^{high}, CD81^{low} DNA^{high}, or CD81^{high} DNA^{low} and CD81^{low} DNA^{low}. As is, I would be forced to conclude that much of the DNA is in CD81^{low} or negative structures and thus is probably not in exosomes. Perhaps autophagy associated membrane compartments are being released by the T cells. Nonetheless, it is clearly in material being released from the T cells and having an impact on DC. Release of such material in vivo could potentially activate DC. I don't feel that STORM is necessary or even helpful in this case. Otherwise I'm happy with the revision.

Reviewer #3 (Remarks to the Author):

There is a loss in T cell viability in the experiments in Fig 6 and Fig S5 that corresponds with the increased numbers of DC that are infected with Vaccinia virus. Since only a small percentage of the DC are infected in their system (<5%, Fig6D), a drop in T cell viability, even of ~13%, or T cell function could account for the increase in infected DC as opposed to release of exosomes. The Authors have performed a lot of work but have not ruled out off-target effects of manumycin.

Replies to Reviewers Comments

Reviewer #1 (Remarks to the Author):

The data in Figure S2 was exactly what I was hoping for regarding single particle analysis. The data suggests that all CD81+ structures are TSG101+ and that all DNA+ structures also have some level of CD81, but they may be somewhat anti-correlated in terms of brightness. A plot of single particle fluorescent signals for CD81 vs TSG101 and CD81 vs DNA would provide sufficient information for anyone trying to reproduce this work a key analytical tool to determine if they have prepared the material correctly. Thus, I would only propose that the authors provide a scatter plot of >100 vesicles, which should be minimally sufficient to demonstrate the frequency of particles that are CD81high DNAhigh, CD81low DNA high, or CD81high DNAlow and CD81low DNAlow. As is I would be forced to conclude that much of the DNA is in CD81low or negative structures and thus is probably not in exosomes. Perhaps autophagy associated membrane compartments are being released by the T cells.

Nonetheless, it's clearly in material being released from the T cells and having an impact on DC. Release of such material in vivo could potentially activate DC. I don't feel that STORM is necessary or even helpful in this case.

Otherwise I'm happy with the revision.

Response: We thank the reviewer for the positive comments. We have now analyzed the images from TRIF microscope and included scatter plot representations for CD81:Tsg101 and CD81:DNA correlation. Spearman coefficient for correlation have been included (0.67 and 0.22, respectively, with $P < 0.001$; $n = 5937$ particles from two different experiments). Data have been included in Suppl Figure 1b.

Reviewer #3 (Remarks to the Author):

There is a loss in T cell viability in the experiments in Fig 6 and Fig S5 that corresponds with the increased numbers of DC that are infected with Vaccinia virus. Since only a small percentage of the DC are infected in their system (<5%, Fig6D), a drop in T cell viability, even of ~13%, or T cell function could account for the increase in infected DC as opposed to release of exosomes. The Authors have performed a lot of work but have not ruled out off-target effects of manumycin.

Response: We have now discussed the possible mechanisms underlying the inhibitory effect of manumycin on exosome production, either by neutral sphingomyelinase or by ras-farnesylase inhibition. Both mechanisms can contribute. Both issues, the risk of off-target effects through ras-farnesylase or IKK-beta inhibition as well as the potential influence of the cell death of nearly 13% of T cells present in the assay vs 87% of exosome-producing, live T cells have also been included in a new paragraph at Discussion.